# Bidirectional pilus processing in the Tad pilus system motor CpaF

**Michael Hohl** [ORCID][1], **Emma J. Banks** [ORCID][2], **Max P. Manley** [ORCID][1], **Tung B. K. Le** [ORCID][2] & **Harry H. Low** [ORCID][1] ✉

The bacterial tight adherence pilus system (TadPS) assembles surface pili essential for adhesion and colonisation in many human pathogens. Pilus dynamics are powered by the ATPase CpaF (TadA), which drives extension and retraction cycles in *Caulobacter crescentus* through an unknown mechanism. Here we use cryogenic electron microscopy and cell-based light microscopy to characterise CpaF mechanism. We show that CpaF assembles into a hexamer with C2 symmetry in different nucleotide states. Nucleotide cycling occurs through an intra-subunit clamp-like mechanism that promotes sequential conformational changes between subunits. Moreover, a comparison of the active sites with different nucleotides bound suggests a mechanism for bidirectional motion. Conserved CpaF residues, predicted to interact with platform proteins CpaG (TadB) and CpaH (TadC), are mutated in vivo to establish their role in pilus processing. Our findings provide a model for how CpaF drives TadPS pilus dynamics and have broad implications for how other ancient type 4 filament family members power pilus assembly.

Tight adherence pilus system (TadPS) pili are filaments that extend into the external milieu and facilitate the anchoring of bacteria to surfaces[1]. By mediating cell adhesion, they represent essential colonisation factors for pathogens such as *Aggregatibacter actinomycetemcomitans*, *Pseudomonas aeruginosa* and *Vibrio vulnificus*[2–4]. In non-pathogenic *Caulobacter crescentus*, anchoring to surfaces occurs through dynamic cycles of pilus extension and retraction until surface contact is established[5,6]. Pilus retraction orients the cells and stimulates the synthesis of the permanent holdfast. In addition, TadPS pili bundle to form biofilms[4,7], facilitate DNA uptake[8] and mediate contact-dependent prey killing in predatory social bacteria[9,10]. The TadPS therefore has a broad functional repertoire conserved across many Gram-negative and Gram-positive bacteria[11]. Despite this prevalence amongst the Eubacteria, the TadPS has an intriguing archaeal heritage as part of the wider type 4 filament family with an ancient lineage originating with the last universal common ancestor (LUCA)[11,12]. Closest archaeal ancestors include the rotary archaellum[13], and those systems producing UV-inducible pili (Ups)[13], archaeal adhesive pili (Aap)[14] and EppA-dependent (Epd) pili[15]. For these pilus systems, little

is known about the secretion system powering pilus assembly although it is likely they share close mechanistic principles with the TadPS given their common evolutionary heritage[16]. Other eubacterial members of the type 4 filament family include the type 4 pilus system[17] that, like the TadPS, assembles retractable surface pili, and the type II secretion system[18] (T2SS), which assembles a dynamic endopilus retained within the periplasmic space.

The typical TadPS constitutes 12-14 genes in a single operon. These include the outer membrane secretin RcpA with its cognate pilotin TadD[19,20] forming a channel through which the assembled Flp1 pilus[21,22] threads and exits the cell. Pilus assembly occurs at the inner membrane and incorporates the most conserved components of the TadPS including the cytoplasmic ATPase TadA[23] and the inner membrane platform proteins TadB and TadC, which share homology to GspF and PilC in the T2SS and type 4 pilus systems, respectively[24]. These three proteins are likely sufficient to drive pilus assembly given that they constitute the minimal TadPS machinery observed in Gram-positive bacteria[4] along with the major pilin Flp1 and TadZ[25] implicated in TadPS positioning within the cell. Whilst TadA is broadly distributed

[1]Department of Infectious Disease, Imperial College, London, UK. [2]Department of Molecular Microbiology, John Innes Centre, Norwich, UK.
✉e-mail: h.low@imperial.ac.uk

across α/β/γ/δ-Proteobacteria, Actinobacteria and Firmicutes[26], it is most closely related to TrbB/VirB11-type ATPases, which comprise part of the wider family of motor ATPases that energise type IV secretion systems (T4SS)[24]. Little is known about how TadA operates including its architecture, mode of assembly and nucleotide cycling mechanism.

In contrast, *Geobacter metallireducens* PilB from the type 4 pilus system is one of the most studied motors. It forms a C2 symmetric hexamer where nucleotide turnover is coupled to a clockwise sequence of symmetric conformational changes dedicated to pilus extension[27]. PilC is predicted to nestle within the central channel of PilB possibly rotating like a drive shaft and transducing lever-like conformational changes that elevate pilins from the inner membrane into the assembling pilus[27]. Intriguingly, by adjusting the conformation of the different nucleotide binding pockets and their relative affinity for different nucleotide states, PilB from *Thermus thermophilus* likely supports counter-clockwise motion[28]. Other mechanisms for PilB- and PilC-mediated pilus assembly have been theorised including a compression model where pilins are extruded from the inner membrane[17] or actively extracted by a trimeric PilC chuck-like arrangement[29]. In the TadPS, evolutionary studies show that TadB and TadC arose from the fission of a single GspF/PilC ancestral gene[11], so that TadB and TadC likely constitute a single functional platform protein. Given the evolutionary homology between the TadPS, T2SS and type 4 pilus system, it is quite possible that they all share conserved mechanisms for pilus assembly mediated by their respective ATPase motors and platform proteins. How pilus retraction occurs, though, is an open question, as most systems do not have dedicated retraction ATPases. This is consistent with primordial type 4 filament systems functioning with just a single ATPase. Type 4 pilus systems are exceptional though as retraction is coordinated by the dedicated ATPase PilT, which was acquired relatively late in the evolutionary process[12,16]. PilT has been extensively crystallised in multiple different conformations and symmetries[30–32]. However, cryogenic electron microscopy (cryo-EM) studies of *G. metallireducens* PilT support a functional C2 symmetric form where counter-clockwise motion antagonises PilB extension resulting in pilus retraction[33].

In the *C. crescentus* TadPS, pilus extension and retraction are mediated by the ATPase CpaF (TadA) alone[26] – yet the mechanism for how one motor drives bidirectional pilus cycling remains elusive. It is also unknown whether this motor bifunctionality is conserved across the TadPS family or is distinct to *C. crescentus*. Similarly, fundamental questions regarding the TadPS pilus assembly mechanism remain given that only the outer membrane secretin and pilotin[20], and the pilus[21], have been structurally characterised from the core apparatus.

Here we use a combination of cryo-EM, light microscopy and cell-based assays to show how CpaF from *C. crescentus* forms a C2 symmetric hexamer primed to couple nucleotide cycling to a clockwise rotational mechanism consistent with right-handed pilus extension. We use sequence conservation analysis, Alphafold modelling and structure-guided mutagenesis to probe how CpaF may engage the platform proteins CpaG (TadB) and CpaH (TadC) and drive pilus assembly. Finally, by comparison of CpaF structures in different nucleotide states and with other related motors, a mechanism is suggested for how CpaF drives both pilus extension and retraction.

## Results

### Cryo-EM structure determination of *C. crescentus* CpaF with AMPPNP

To probe the mechanism by which CpaF powers the TadPS, single particle cryo-EM was undertaken. CpaF from *C. crescentus* was cloned and purified to homogeneity (Supplementary Fig. 1A). After incubation with the non-hydrolysable ATP analogue AMPPNP and in the presence of Mg$^{2+}$, the sample was vitrified (Supplementary Fig. 1A) using holey grids and a data set collected for high resolution structure determination. Subsequent data processing yielded a map constituting six

subunits arranged in a C2 symmetric ring (Supplementary Fig. 1B and 1E, and Supplementary Table 1). Side chain detail was sufficient to build amino acid residues 150–502 relating to the N-terminal domain 2 (NTD2) and C-terminal domain (CTD) for each subunit at 3.8 Å resolution overall (Supplementary Fig. 1C, 1D and 2A). Whilst residues 1–79 termed N-terminal domain 0 (NTD0) were predicted to be disordered and were not observed in the map, residues 80–149 constituted the N-terminal domain 1 (NTD1) and were predicted to form a 3-helix motif. Due to flexibility between the NTD1 and NTD2, map quality was insufficient for NTD1 model building. A focussed refinement strategy zoned on one subunit improved the resolution of the NTD1 map (Supplementary Fig. 2B and 2C) so that an Alphafold[34] model of the 3-helix motif could be rigid body fitted and modelled (Supplementary Fig. 1D). Henceforth, this CpaF structure obtained in the presence of AMPPNP is termed CpaF$_{AMPPNP}$.

### The CpaF monomer has a modular fold similar to FlaI-like archaeal motors

The basic building block (asymmetric unit) of the CpaF oligomer is a trimer that repeats to form an oval-shaped hexamer with C2 symmetry -128 Å wide and -80 Å high (Fig. 1A). Within each subunit, the helical NTD1 is connected to the NTD2 by a short linker (Hinge 1) with conserved residues G147 and G149 likely mediating flexibility between these domains (Fig. 1B and Supplementary Fig. 3). The NTD2 comprises a layer of six β-sheets with two helices and connects to the CTD via an eight-residue lysine-rich linker (Hinge 2) that wraps around and forms part of the nucleotide-binding pocket. The CTD has a conserved RecA-like fold[35] with central β-sheets surrounded by α-helices on each side. It includes classical motifs such as P-loop, Walker A, Walker B, ASP box and HIS box (Supplementary Fig. 3) with the nucleotide binding pocket cradled between CTD and NTD2 interfaces. Hidden Markov Model analysis shows that the overall fold of the CpaF monomer is like the archaeal motor FlaI in *Sulfolobus acidocaldarius*, which powers archaellum rotation[36], and the FlaI-like ATPase GspE2 in *Archaeoglobus fulgidus* with unknown function[37]. Comparison of their structures provides firm evidence for their shared archaeal ancestry based on tertiary fold, consistent with evolutionary sequence analyses (Supplementary Fig. 4)[11].

### CpaF$_{AMPPNP}$ subunits pack in distinct conformations that induce C2 symmetry within the hexamer

To assemble the CpaF hexameric ring, subunits associate laterally around an extensive central cavity with the CTDs forming the ring base, the NTD2s stacked above and the NTD1s protruding at the top. Importantly, the NTD2s are rotationally offset so that the NTD2 of subunit n interfaces with the CTD of subunit $n+1$ in a domain swap (Fig. 1A). This structural unit is henceforth termed NTD2/CTD$_{n+1}$. Although CpaF was incubated with AMPPNP, this nucleotide was symmetrically bound in only two of the nucleotide-binding pockets within the hexamer. ADP coordinated to a Mg$^{2+}$ ion (ADP/Mg$^{2+}$) was symmetrically bound in the neighbouring subunit, with ADP alone in the remaining pair (Fig. 1A). The ADP is likely derived from AMPPNP hydrolysis given non-hydrolysable ATP analogues are not entirely resistant to hydrolysis[38]. Mg$^{2+}$ was present in the vitrification buffer indicating a requirement for specific active site geometry to support its recruitment. Comparison of the three subunits in the asymmetric unit showed that whilst the AMPPNP and ADP bound chains were relatively similar with RMSD Cα = 1.2 Å, the ADP/Mg$^{2+}$ chain underwent a significant conformational change where the CTD rotated -34˚ around Hinge 2. This induced the CTD to swing upwards 18 Å relative to the NTD2 resulting in a clamp-like effect around the nucleotide binding pocket (Fig. 1C and Supplementary Movie 1). Therefore, it is largely the packing mismatch of the ADP/Mg$^{2+}$ bound subunits relative to the other four that generates the C2 symmetry within the hexamer. Given this, it is important to note that the superposition of the three distinct

                                                        

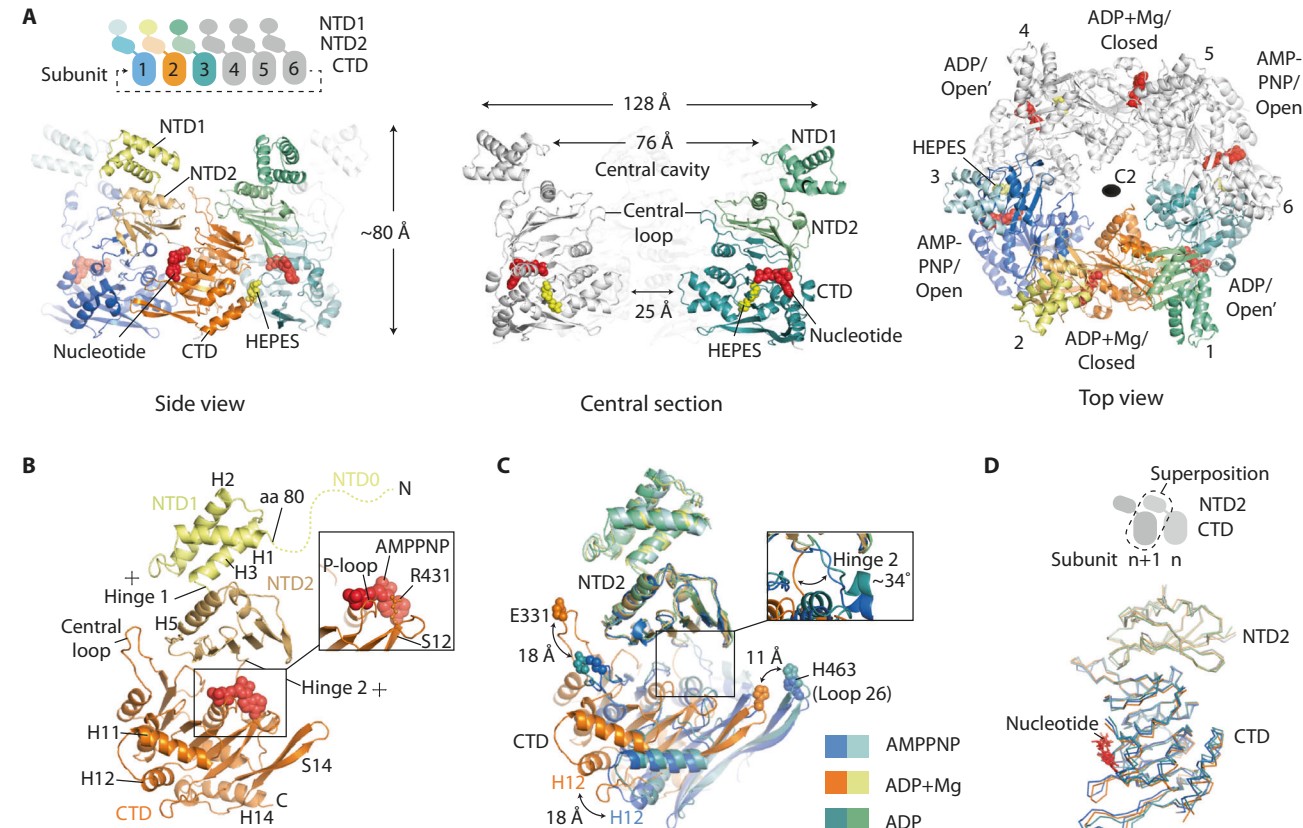

**Fig. 1 | Cryo-EM structure of *C. crescentus* CpaF$_{AMPPNP}$. A** Model of the C2 symmetric CpaF hexamer with subunits binding AMPPNP, ADP/Mg$^{2+}$ or ADP. **B** The CpaF monomer bound to AMPPNP with key nomenclature. Flexibility is observed in Hinge 1 between NTD1 and NTD2, and in Hinge 2 between NTD2 and CTD. **C** Superposition of the three subunits in the asymmetric unit clamp-like conformational change mediated by Hinge 2. For clarity, ligands have been omitted. Residues E331 (Central Loop) and H463 (Loop 26) were highlighted to showcase CTD movement. **D** Superposition of the three distinct NTD2/CTD$_{n+1}$ units within each CpaF hexamer showing close similarity. For each subunit, the NTD2 of subunit n interfaces with the CTD of subunit $n+1$ in a domain swap.

NTD2/CTD$_{n+1}$ units within the hexamer shows close similarity with RMSD Cα ≤ 1.1 Å (Fig. 1D). This means that the conformational changes observed between subunits within the CpaF hexamer are predominantly mediated by large-scale rigid body shifts between NTD2/CTD$_{n+1}$ units, with Hinge 2 providing the necessary flexibility.

### The CpaF$_{AMPPNP}$ hexamer exhibits symmetric open, closed and open′ nucleotide binding pockets that reveal the catalytic cycle

The conformation of the three subunits in the asymmetric unit, and how they interact with their neighbours defines the shape and accessibility of their nucleotide binding pockets. In those subunits where AMPPNP and ADP alone are bound, the active sites are in similar but still distinct conformations termed open and open′, respectively, with the solvent channel leading to the nucleotide-binding pocket sufficient to facilitate the passage of nucleotide (Figs. 1A, 2A and 2B). These open/open′ states are also characterised by a solvent-accessible cleft running between the CTD and the CTD$_{n+1}$ of the neighbouring subunit (termed henceforth the CTD/CTD$_{n+1}$ cleft). The cleft is broad enough to support the binding of a HEPES moiety recruited during sample purification (Figs. 2A, B). The equivalent cleft in PilT has been shown to bind ethylene glycol[33]. In the ADP/Mg$^{2+}$-bound subunit sandwiched between the open/open′ state subunits, the active site is in a closed conformation with the nucleotide-binding pocket largely occluded by the clamped CTD and NTD2. Consequently, the CTD/CTD$_{n+1}$ cleft is compressed with no HEPES bound, and Loop 26 between β-sheets 14 and 15 is now directly contacting H13 in CTD$_{n+1}$ (Fig. 2A).

Close analysis of the active sites reveals how nucleotide cycling and catalysis are achieved. In the open conformation, the adenine moiety of the bound AMPPNP is secured at its side and back by conserved R431 in β-sheet 13 and non-conserved F256, respectively (Figs. 1B and 2A). K244 is 4.0 Å from the ribose so within range for hydrogen bonding (Fig. 2A and Supplementary Fig. 5). Note that a K244R mutation (in combination with F243L) reduces pilus extension and retraction rate[26] possibly by modulating nucleotide access to the binding pocket or impeding its binding. The rest of the nucleotide is cradled by the P-loop including K287 within the conserved consensus sequence coordinated to the γ-phosphate (Fig. 2C and Supplementary Fig. 5). Conserved ASP box E312, Walker B E357, R359 and HIS box H382 are 4-6 Å from the γ-phosphate tip. NTD2 residues R217 and conserved R223 located within β-sheets 4 and 5, respectively, are turned over 12 Å away from the β- and γ-phosphates. Similarly, R347 in Helix 9 from the neighbouring CTD$_{n+1}$ is over 12 Å away from the γ-phosphate. Here, R347 and R359 are some of the most conserved residues amongst CpaF homologues (Supplementary Fig. 3) and across other type 4 filament family motors including the archaellum, T2SS and type 4 pilus system (Supplementary Fig. 6). Nucleotide binding is therefore stabilised by a ring of basic charged residues. However, without R217, R223 and R347 engaged with the phosphates, the active site in the open state is not primed for catalysis. In contrast, the neighbouring subunit in the closed conformation has undergone significant rearrangement and is in a catalytically active state (Fig. 2C and Supplementary Fig. 5). Map was observed for the water-activating Mg$^{2+}$ ion coordinated with ASP box E312 and Walker B E357 located 4 Å and 5 Å away, respectively; whilst R217 and R223 are now stabilising the ADP β-phosphate positioned just 3 Å and 4 Å apart, respectively. Similarly, R347 from the neighbouring CTD$_{n+1}$ has moved less than 7 Å away from the

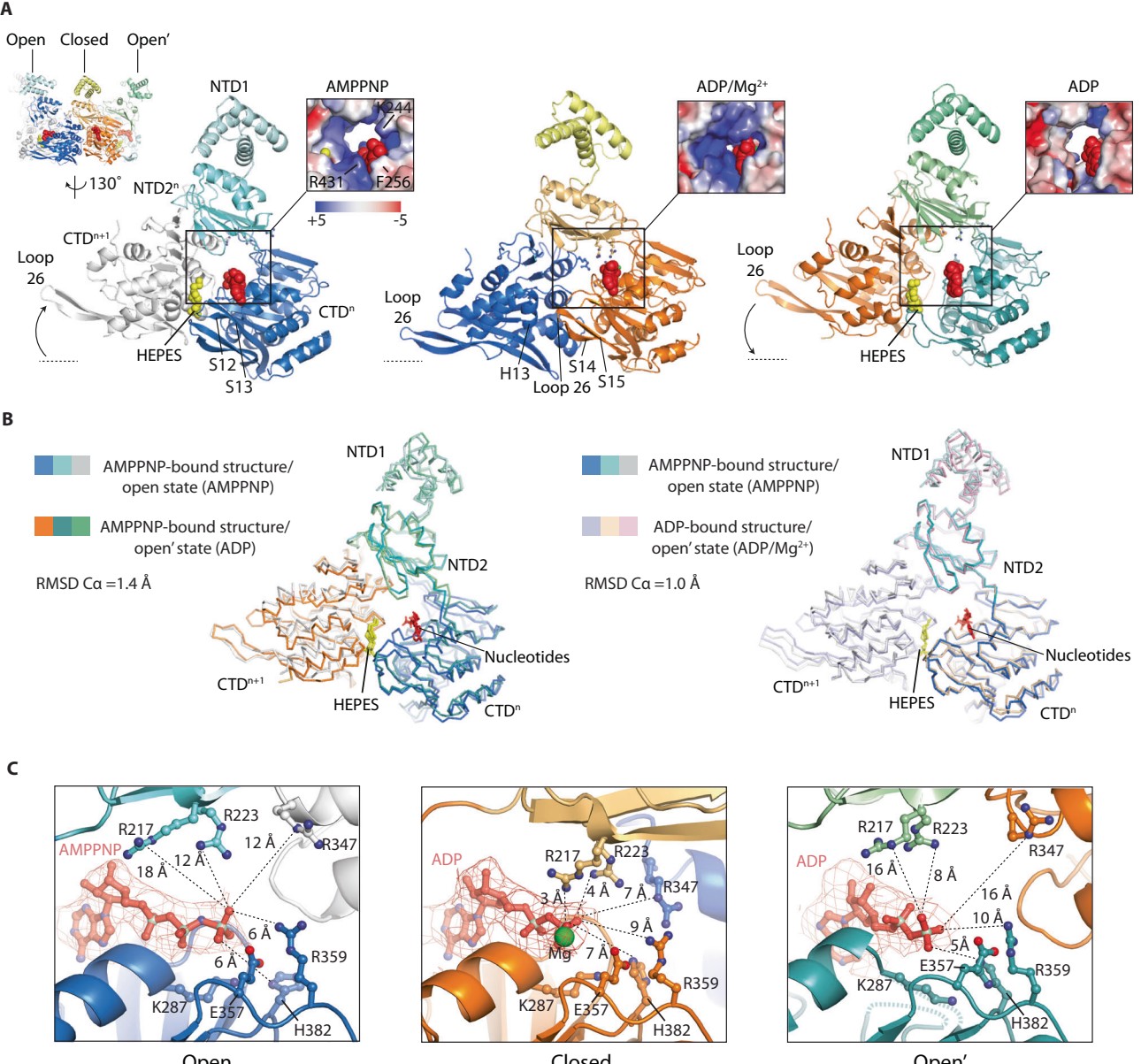

**Fig. 2 | CpaF$_{AMPPNP}$ has symmetric open, closed and open' nucleotide binding pockets that showcase the catalytic cycle. A** The three subunits in the asymmetric unit bind AMPPNP, ADP/Mg$^{2+}$ or ADP, constituting open, closed and open' active site states, respectively. Each active site is formed by the NTD2 and CTD of subunit n (NTD2$_n$ and CTD$_n$), and the CTD of subunit $n+1$ (CTD$_{n+1}$). Zoom window shows surface rendered to electrostatic charge. Blue to red spectrum represent positive to negative charges with units $k_BT/e_c$. **B** Superposition of subunits within the asymmetric unit with their neighbouring CTD$_{n+1}$. (Left) Comparison of

CpaF$_{AMPPNP}$ open and open' states. (Right) Comparison of CpaF$_{AMPPNP}$ open and CpaF$_{ADP}$ open' states. CpaF$_{AMPPNP}$ subunits follow the colour code used in (**A**). **C** Analysis of open, closed and open' active site states with AMPPNP, ADP/Mg$^{2+}$ or ADP alone bound, respectively. The open' state has a relatively disordered active site with an unstructured P-loop. For clarity, ASP Box E312 has been omitted as well as the distances between E357 and the nucleotide terminal phosphate. Both residues with respective distances are shown in Supplementary Fig. 5.

β-phosphate and is positioned to stabilise the γ-phosphate during the ATP hydrolysis transition state. In the archaeal FlaI, the equivalent residue R326 has been shown to undertake this stabilising role locating within the hydrogen bonding distance of the released phosphate positioned 4.2 Å from the β-phosphate[36]. HIS box H382 has switched rotamer moving into a position where it may coordinate the γ-phosphate, similar to the role of HIS box H420 in PilB[27]. In CpaF, a channel connecting the nucleotide-binding pocket to the central cavity of the hexameric ring is sufficiently broad to enable the γ-phosphate to exit the active site, which likely explains its absence. Finally, examination of the neighbouring open' conformation subunit with ADP alone bound reveals an active site with similarities to the open state (Fig. 2B and

Supplementary Fig. 5) with R217, R223 and R347 rotated away from the nucleotide (Fig. 2C). However, of notable difference, the ADP ligand is loosely bound with a relatively disordered active site including unstructured P-loop with no model built from G282 to G286. Importantly, the sidechain of highly conserved P-loop K287, which is critical for nucleotide binding, has partially disengaged from the nucleotide with HIS box residue H382 moved inwards and now just 5 Å from the β-phosphate. Similarly, β-sheets 12 and 13 become partially disordered (no model built for residues 425–431) so that their connecting loop no longer contacts the HEPES moiety in the CTD/CTD$_{n+1}$ cleft as in the open conformation (Fig. 2A). This disordered region includes R431, which cannot now effectively secure the adenine moiety of the

nucleotide in the active site. Overall, the architecture of the open conformation is consistent with a state where the ADP nucleotide is bound with low affinity and primed for release. In summary, the $CpaF_{AMPPNP}$ structure reveals an asymmetric unit in three conformations. Importantly, these conformations showcase the key stages in the nucleotide hydrolysis cycle including ATP loading (open state), ATP hydrolysis to ADP with the exit of the γ-phosphate from the active site (closed state), and ADP bound with low affinity primed for release (open' state).

### In the presence of ADP, the CpaF open' state resembles the $CpaF_{AMPPNP}$ open state

To understand how different nucleotide states might regulate CpaF assembly and conformation, the structure of CpaF was determined by cryo-EM in the presence of ADP and $Mg^{2+}$. CpaF incubated with ADP yielded our highest quality map at 3.1 Å resolution overall facilitating a build of the NTD2 and CTD with excellent sidechain detail for all subunits (Supplementary Fig. 1E, 2D and 2F, and Supplementary Table 1). The 3-helix bundle of the NTD1 was again poorly ordered indicating domain flexibility relative to NTD2. The NTD1 from the AMPNP-bound model was therefore rigid-body fitted in the unsharpened map and modelled to complete the build. Henceforth, this CpaF structure obtained in the presence of ADP is termed $CpaF_{ADP}$.

$CpaF_{ADP}$ formed a C2 symmetric hexamer with dimensions and subunit packing similar to $CpaF_{AMPPNP}$ (Fig. 3A). Superposition of the three distinct $NTD2/CTD_{n+1}$ units within the $CpaF_{ADP}$ hexamer showed remarkable alignment with RMSD Cα ≤ 0.35 Å (Fig. 3B). Similarly, superposition of the asymmetric units from both $CpaF_{ADP}$ and $CpaF_{AMPPNP}$ showed near identical packing between the $NTD2/CTD_{n+1}$ units with RMSD Cα = 1.5 Å (Fig. 3C). As observed in $CpaF_{AMPPNP}$, subunits and their nucleotide-binding pockets were in symmetric open, closed and open conformations. However, ADP/ $Mg^{2+}$ was now bound in each nucleotide-binding pocket (Fig. 3D and Supplementary Fig. 5). Analysis of the active sites revealed an arrangement and geometry of key residues that was similar to $CpaF_{AMPPNP}$. Specifically, in the open and open' states, residues R217 and R223, and R347 from $CTD_{n+1}$, were all positioned > 15 Å from the nucleotide and unable to engage it. In the open state, K287, E357, R359 and H382 were positioned like in the $CpaF_{AMPPNP}$ open state despite the absence of the γ-phosphate. ASP box E312 was coordinated 4 Å from the $Mg^{2+}$. In the closed conformation, R217 and R223 were positioned less than 5 Å and 3 Å away from the β-phosphate, whilst R347 from $CTD_{n+1}$ was now just 8 Å away. All these residues were suitably positioned to stabilise the transition state during nucleotide hydrolysis. However, despite these similarities, there were critical differences between the open' conformations of $CpaF_{ADP}$ and $CpaF_{AMPPNP}$. In $CpaF_{ADP}$, the map around the nucleotide binding pocket was now fully ordered with the P-loop well resolved. Crucially, HIS box residue H382 was turned away from the nucleotide with P-loop K287 fully engaging the β-phosphate just like in $CpaF_{AMPPNP}$ open state. Moreover, superposition of subunits for $CpaF_{ADP}$ open' state with $CpaF_{AMPPNP}$ open state, including their interfaced $CTD_{n+1}$, revealed remarkable similarity with RMSD Cα = 1.0 Å (Fig. 2B). These two states therefore share closer similarity than the open and open' states of $CpaF_{AMPPNP}$ with an RMSD Cα = 1.4 Å when superposed (Fig. 2B). Broadly, this means that when ADP/$Mg^{2+}$ is bound to $CpaF_{ADP}$ in the open state (Fig. 3D), then the nucleotide binding pocket of the open' state subunit resides in a conformation similar to the open state of $CpaF_{AMPPNP}$ (Fig. 2C and Supplementary Fig. 5). This finding suggests a model where the $CpaF_{ADP}$ open' state has the appropriate conformation and geometry of key residues to support AMPPNP (or ATP) binding when ADP/$Mg^{2+}$ is bound in the open state subunit.

### CpaF assembles as a relaxed C2 symmetric hexamer in the absence of nucleotide

To understand how the absence of nucleotide might regulate CpaF assembly and conformation, the structure of CpaF was determined by cryo-EM without nucleotide addition but in the presence of $Mg^{2+}$. In this case, CpaF was less stable yielding a map resolved to 4.0 Å resolution overall (Supplementary Figs. 1E, 2E and 2G, and Supplementary Table 1). Map quality was sufficient to build complete models of the CTD and NTD2 for all subunits (Supplementary Fig. 2G). Again, the NTD1s were poorly ordered in the map and were built using the NTD1 from $CpaF_{AMPPNP}$ rigid-body fitted and modelled. Henceforth, this CpaF structure obtained in the absence of nucleotide is termed $CpaF_{APO}$.

$CpaF_{APO}$ formed a C2 symmetric hexamer with subunits arranged in an open, closed and open' conformation like $CpaF_{AMPPNP}$ and $CpaF_{ADP}$ (Supplementary Fig. 7A). No nucleotide or $Mg^{2+}$ was observed in the nucleotide-binding pockets (Supplementary Fig. 7B) although HEPES was bound in the $CTD/CTD_{n+1}$ clefts in both open and open' state subunits. Superposition of the three distinct $NTD2/CTD_{n+1}$ units within the $CpaF_{APO}$ hexamer showed close alignment with RMSD Cα ≤ 0.7 Å (Supplementary Fig. 7C). However, comparison of the $CpaF_{APO}$ and $CpaF_{ADP}$ asymmetric units showed that whilst similar, they do not pack identically and cannot be considered equivalent to $CpaF_{ADP}$ superposed on $CpaF_{AMPPNP}$. Specifically, Hinge 2 within the open state subunit is straightened so that its NTD2 moves towards the neighbouring NTD2 of the closed subunit (Fig. 3C). Consequently, the nucleotide-binding pocket in the open state subunit differs significantly in comparison to $CpaF_{AMPPNP}$ and $CpaF_{ADP}$ as the associated $NTD2/CTD_{n+1}$ unit has rotated (Supplementary Fig. 7B). The overall effect is a relaxed and more rounded C2 symmetric hexamer. $CpaF_{APO}$ is therefore consistent with a model where CpaF preferentially assembles as a C2 symmetric hexamer irrespective of nucleotide state. Critically though, nucleotide binding accentuates the oval-shaped C2 symmetry observed in $CpaF_{AMPPNP}$ and $CpaF_{ADP}$ hexamers and switches the conformation of the open state nucleotide binding pocket, with this form likely to be prevalent in vivo given the high nucleotide concentration in the cell.

### CpaF Loop 8 and Central Loop likely interact with CpaG and CpaH, and are essential for TadPS function

To investigate CpaF mechanism, $CpaF_{AMPPNP}$ and $CpaF_{ADP}$ structures combined with sequence alignments (Supplementary Figs. 3 and 6), were used to select truncations or highly conserved residues (Fig. 4A) for chromosomal mutation. The impact of these mutations was assessed in two cell-based in vivo functional assays. The first was a TadPS pilus-dependent phage infection using ΦCbK[39] where TadPS deactivation impairs phage entry, cell lysis and ultimately plaque formation in *C. crescentus* lawns. However, this assay does not distinguish extension versus retraction mutants as pilus retraction is not essential for ΦCbK infection[26]. The second fluorescence microscopy-based assay was therefore performed to directly visualise and quantify TadPS surface pili, which are produced by *C. crescentus* swarmer cells. This assay involves fluorescent labelling of Tad pili by mutating T36C within the major pilin protein PilA (Flp1). This mutation facilitates binding of a thiol-reactive maleimide-AF488 dye (which has strong affinity for cysteine residues) to the pili without compromising pilus integrity[6]. Addition of dye to cell culture results in direct labelling of extended cell-surface pili and also the cell body due to retraction of fluorescent pilin subunits into the cell envelope. Pilus extension mutants therefore have no fluorescent pili or cell bodies, whereas pilus retraction mutants have fluorescent pili but dark cell bodies[26].

Working with the ΦCbK assay first, we verified that the PilA T36C strain functioned like wildtype (Fig. 4B)[6]. CpaF was then deleted (PilA T36C Δ*cpaF*) resulting in loss of phage infection before

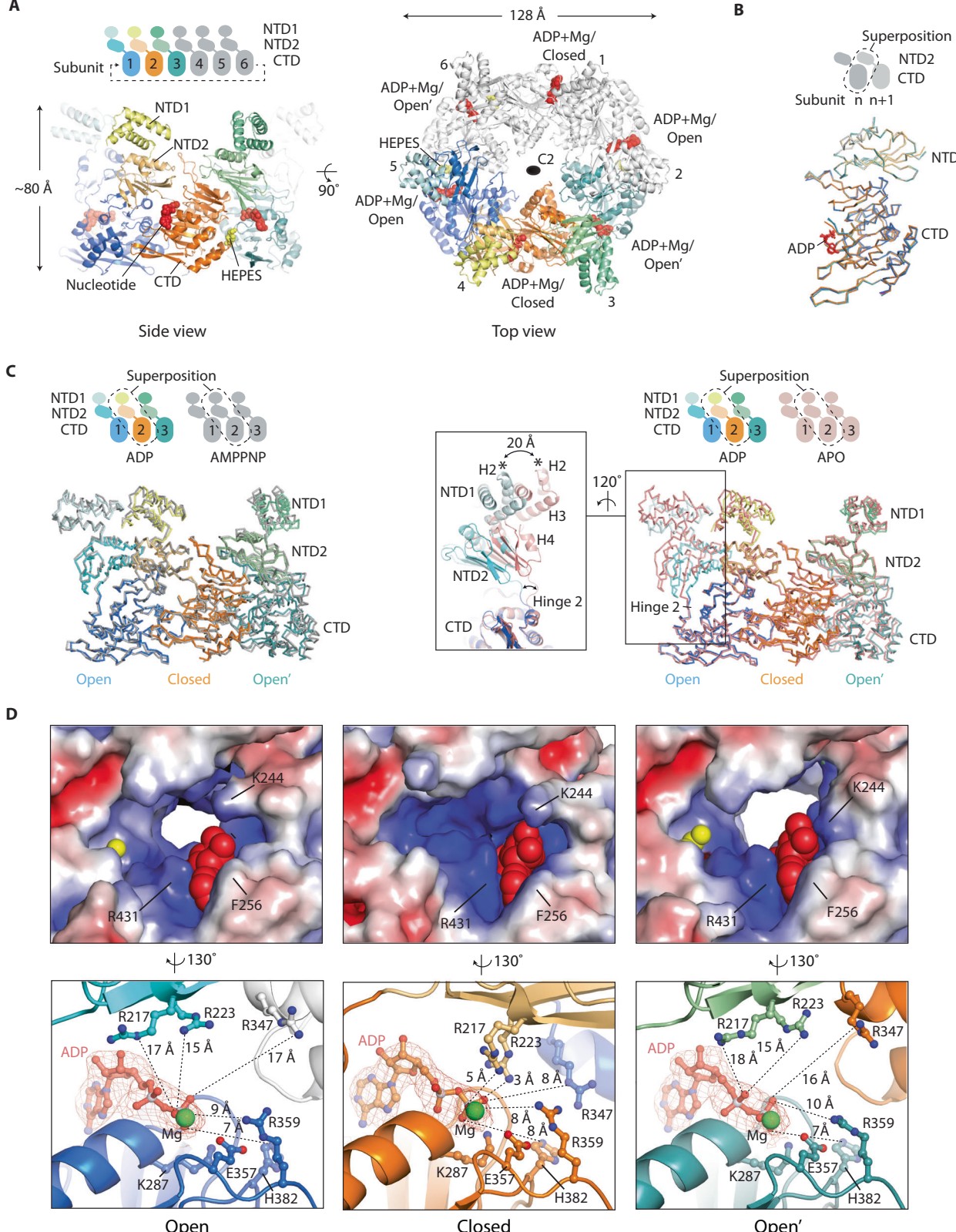

**Fig. 3 | Cryo-EM structure of CpaF_ADP. A** Model of the C2 symmetric CpaF hexamer where all subunits bind ADP/Mg²⁺. **B** Superposition of the three distinct NTD2/CTD_{n+1} units within CpaF_ADP show close similarity. **C** Comparison of CpaF asymmetric units in various nucleotide states. (Left) CpaF_ADP superposed with CpaF_AMPPNP. (Right) CpaF_ADP superposed with CpaF_APO where no nucleotide was observed in the active sites. Zoom box highlights the conformational change in the open state subunit induced by Hinge 2 straightening resulting in rotation of the NTD2 (and NTD1) towards the neighbouring closed subunit. In all superpositions ligands were omitted for clarity. **D** Analysis of open, closed and open' active site states with ADP/Mg²⁺ universally bound. Surface rendering (top row) shows electrostatic charge where blue to red spectrum represent positive to negative charges with units $k_BT/e_c$. For clarity, ASP Box E312 has been omitted as well as the distances between E357 and the nucleotide terminal phosphate (bottom row). Both residues with respective distances are shown in Supplementary Fig. 5.

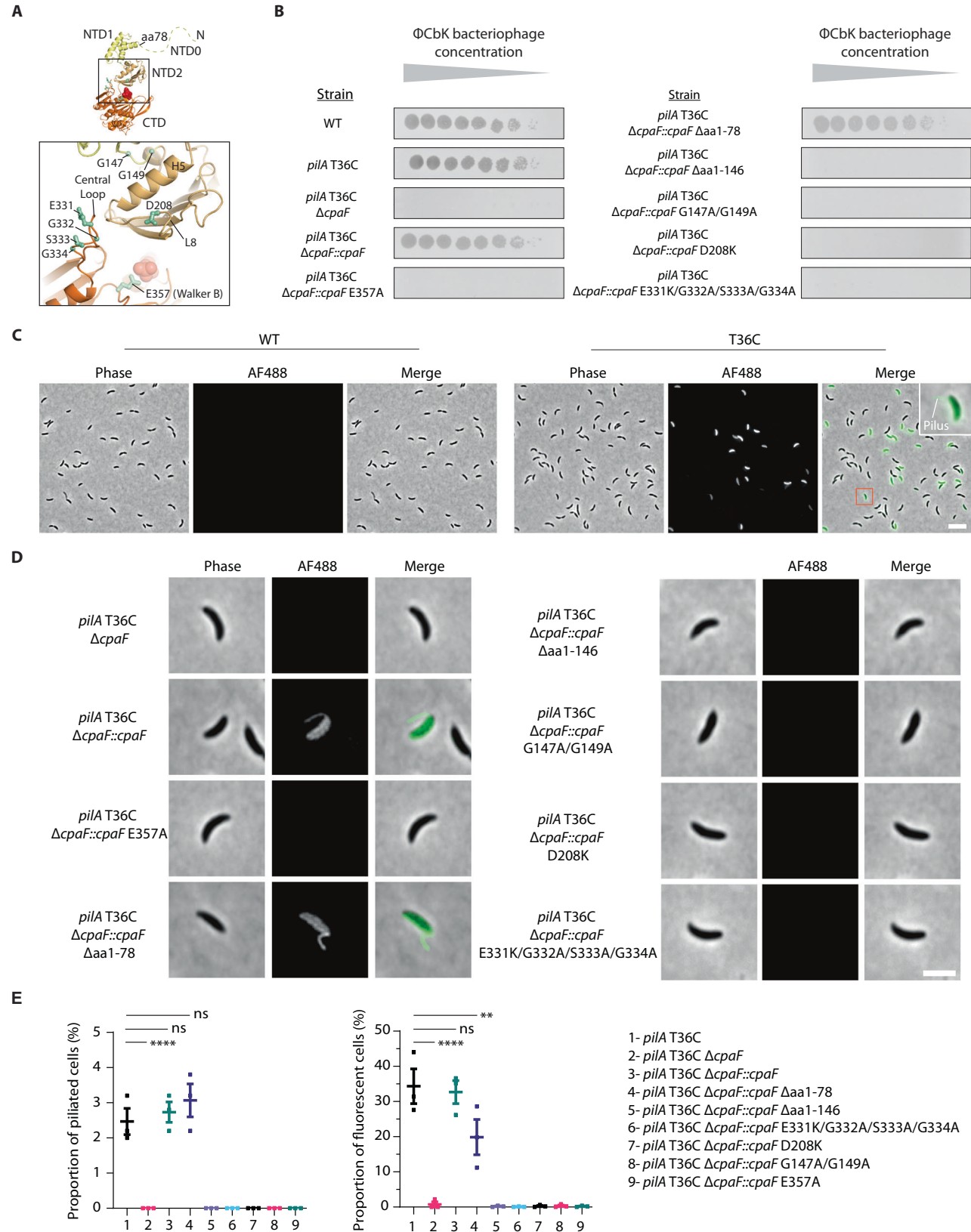

complementation (PilA T36C Δ*cpaF*::*cpaF*) restored infection levels. This validated the assay and showed that CpaF is essential for TadPS function[6,23,40]. All subsequent mutations were constructed in this complemented strain. As a final control, a CpaF Walker B mutant (E357A) was generated resulting in the absence of phage plaques presumably due to loss of CpaF nucleotide hydrolysis. Cellular

expression for individual mutants was verified by Western blot (Supplementary Fig. 8A) whilst hexamer formation was verified by purification and negative stain EM analysis (Supplementary Fig. 8B). The CpaF NTD0 (Supplementary Fig. 3) is predicted to be disordered with unknown function. In order to probe its role, NTD0 was truncated (Δaa1-78) and its ability to support phage infection assayed. Wildtype

**Fig. 4 | Effect of targeted CpaF mutagenesis on TadPS function in vivo.**
**A** Position of truncations or highly conserved residues within the CpaF monomer selected for chromosomal mutation. **B** Phage infection assays using ΦCbK on the mutant strains where TadPS deactivation impairs phage entry, cell lysis and plaque formation in *C. crescentus*. Images are representative of three biological repeats. **C** Demonstration of Tad pili labelling using a fluorescent AF488-conjugated thiol-reactive maleimide dye that is unreactive with wildtype PilA (left) but reacts and decorates pili that have a PilA T36C mutation (right). Inset image shows a magnified *C. crescentus* cell where both a fluorescent pilus (indicating functional extension) and fluorescent cell body (indicating functional retraction) can be observed. Note that only *C. crescentus* swarmer-type cells are capable of producing Tad pili. Without this capability, other cell types (stalked and predivisional cells) do not fluoresce. Scale bar = 5 μm for all panels. Images are representative of three

biological repeats. **D** Direct visualisation of Tad pili by fluorescence microscopy in different *C. crescentus* CpaF mutants as described in (**C**). Scale bars = 2 μm for all panels. Images are representative of three biological repeats. **E** (Left panel) Quantification of piliated cells observed in the CpaF truncations or mutant strains, relating to (**C**) and (**D**). For the proportion of fluorescent cells in the CpaF Δaa1-78 truncation strain $P < 0.0001$ (right panel). The mean of three biological replicates (N = 500 cells counted per biological repeat) was plotted with standard error of the mean, **** $P < 0.0001$, ns = not statistically significant. Statistics were determined using Dunnett's one-way ANOVA test. (Right panel) Quantification of fluorescent cells observed in the CpaF truncations or mutant strains, relating to (**C**) and (**D**). The significant reduction in fluorescent cells in the CpaF Δaa1-78 strain (** $P = 0.002$) indicates a partial retraction mutant. Source data are provided as a Source Data file.

levels of phage plaques were observed indicating no requirement for this motif in pilus extension (Fig. 4B). The NTD0 truncation was expanded to include removal of NTD1 (Δaa1-146), which resulted in loss of phage infection, however, this mutant should be interpreted with caution as in vivo expression was not readily detected by Western blot (Supplementary Fig. 8A) despite purifying and assembling in vitro (Supplementary Fig. 8B). In contrast, the mutation of two conserved glycines (G147A/G149A) positioned within Hinge 1, connecting the NTD1 to NTD2, inhibited phage infection. Therefore, NTD1 flexibility is an essential part of the CpaF mechanism.

During the course of this study, Alphafold-Multimer[41] enabled the modelling of CpaF with CpaG and CpaH (Fig. 5A and Supplementary Fig. 9) showing a heterodimer that bridges two neighbouring CpaF subunits. Specifically, each CpaG or CpaH subunit interacts with Helix 5 conjoined with Loop 8, and the Central Loop of CpaF (Fig. 5B). To test the predicted interface between CpaF and the platform proteins CpaG and CpaH, two mutants were generated. The first mutant D208K resides in Loop 8, which is conjoined to Helix 5 within the NTD2, whereas the second mutant E331K/G332A/S333A/G334A located to the Central Loop within the CTD (Fig. 4A and Supplementary Fig. 3). These residues are highly conserved amongst CpaF homologues and locate to the CpaF surface facing the inner membrane where CpaG and CpaH are positioned[19]. For both these mutants phage infection was inhibited. Whilst it should be cautioned that a proportion of this inhibition may be due to lowered CpaF expression in vivo (Supplementary Fig. 8A) and stability during purification in vitro, the inhibition is consistent with the mutated residues playing an essential role in CpaF function, likely by interfacing with the platform proteins CpaG and CpaH.

To further clarify whether any of the ΦCbK infection-negative mutants were competent to assemble but not retract pili, the mutants were visualised by fluorescence microscopy. As expected, fluorescent cells and pili were observed for the PilA T36C strain (Figs. 4C, E) and the CpaF complemented strain. However, for the CpaF Δaa1-78 truncation strain, whilst the number of piliated cells was similar to the PilA T36C strain, the proportion of fluorescent cells was significantly reduced from 34.3 % ± 5.0 in the PilA T36C strain to 19.9 % ± 5.0 ($P < 0.0001$; Fig. 4E). The CpaF Δaa1-78 truncation strain therefore has a reduced capacity to retract pili. For all other mutants no fluorescent pili or cells were observed confirming a loss of pilus extension rather than a retraction defect (Figs. 4D, E). Collectively, these mutants uncover key residues in CpaF including how the CpaF N-terminus positively regulates pilus retraction. In addition, they highlight the important role of both Loop 8 and the Central Loop in TadPS function likely due to their direct interaction with CpaG and CpaH platform proteins.

## Discussion
Here we use CpaF from *C. crescentus* as a model to probe how one motor might power both pilus extension and retraction in this system and possibly in other bacterial homologues. The CpaF$_{AMPPNP}$ hexamer has subunits in three symmetric conformations with AMPPNP bound in

the open state, ADP/Mg$^{2+}$ in the closed state and ADP in the open' state. The structure therefore represents three key stages in the nucleotide hydrolysis cycle where i) the open state has high affinity for ATP and the catalytic residues in the active site are disengaged; ii) the closed state is catalytically active and captured in a post-hydrolysis state with high affinity for ADP, catalytic residues engaged around the nucleotide, and the γ-phosphate released into the central cavity; iii) the open' state has low affinity for ADP, the catalytic residues are disengaged in the active site and the nucleotide is primed for release. The sequence of these states, combined with the C2 symmetry, is consistent with CpaF$_{AMPPNP}$ geared for nucleotide cycling in a clockwise motion when viewed from the NTD1 side of the hexamer (Fig. 6). In this model, the long axis of the oval-shaped central cavity rotates ~60° with each step of the nucleotide hydrolysis cycle and naturally supports the extension of the right-handed *C. crescentus* PilA pilus[21].

Conversely, one way to achieve pilus retraction would be to promote counter-clockwise motion in the CpaF motor. This could be achieved if the open' state switched to have a high affinity for ATP and the open state to have low affinity for ADP thereby reversing the direction of nucleotide-induced conformational changes over the hydrolysis cycle. In support of this nucleotide switch model, analysis of CpaF$_{ADP}$ open' state shows how the active site geometry may be configured to closely resemble an ATP binding conformation. The CpaF$_{ADP}$ open' state active site is ordered and whilst ADP/Mg$^{2+}$ is bound it shows remarkable similarity in backbone conformation and side chain geometry to the active site of CpaF$_{AMPPNP}$ open state with AMPPNP bound (Fig. 6A and Supplementary Fig. 5). This is in contrast to CpaF$_{AMPPNP}$ open' state which is comparatively disordered with unstructured P-loop (Fig. 2C) and partially unstructured β-sheets 12 and 13 so that R431 no longer secures the nucleotide base (Supplementary Fig. 5). Concurrently, the CpaF$_{ADP}$ open' state shows close similarity to the active site of CpaF$_{ADP}$ open state (Fig. 6A and Supplementary Fig. 5). As the nucleotide binding pockets are solvent accessible in both these states, the CpaF$_{ADP}$ structure is consistent with the nucleotide switch model where ATP and ADP binding in the open' and open states, respectively, would potentiate counter-clockwise motion (Fig. 6B). How a nucleotide switch may be regulated to shift between clockwise and counter-clockwise motion is unclear. Some CpaF homologues have variable NTD0 extensions (Supplementary Fig. 3) with sufficient length to reach the nucleotide binding pockets and in principle to allosterically modulate the geometry of active sites or nucleotide access and affinity. In this respect, it is intriguing that Alphafold models N-terminal Helix 0 (aa 57–66) at the mouth of the nucleotide binding pocket where it could modulate or gate the nucleotide binding pocket (Fig. 5A). Any such interaction would be dynamic as density was not observed for H0 in the CpaF maps. As loss of the NTD0 (aa1–78) reduces pilus retraction (Fig. 4E), this motif has the capacity to control or at least contribute to CpaF pilus extension and retraction bifunctionality. Importantly, the role of the NTD0 in modulating CpaF bifunctionality is also supported by comparison studies between CpaF in both *C. crescentus* and *Asticcacaulis biprosthecum* where differences

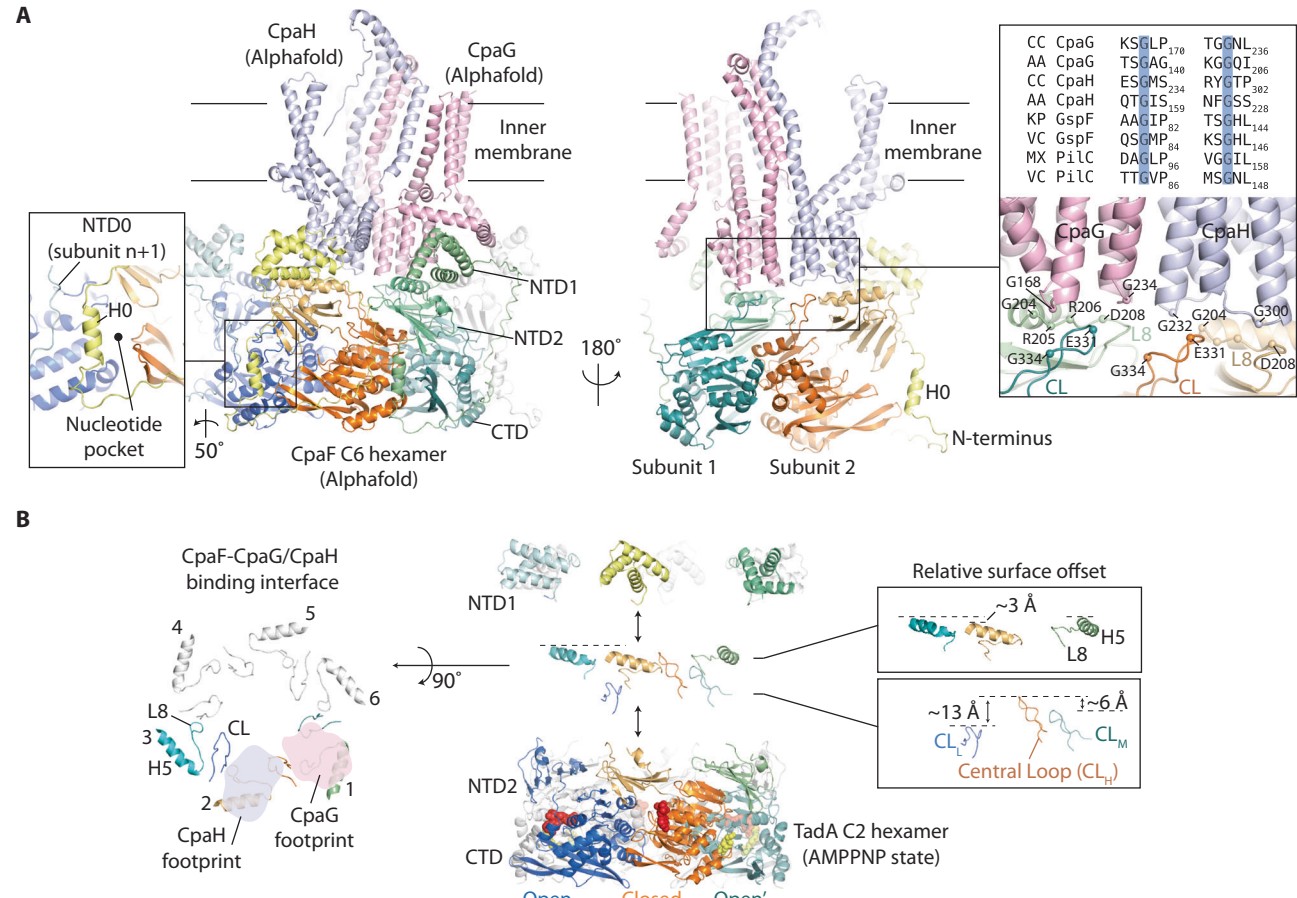

**Fig. 5 | In-silico modelling of CpaF with CpaG/CpaH. A** Alphafold model with CpaF:CpaG:CpaH stoichiometry of 6:3:3, overall pLDDT score = 74.9 (Supplementary Fig. 9). Here, just one CpaG/CpaH heterodimer bound to six CpaF subunits (left panel) and one CpaG/CpaH heterodimer bound to two CpaF subunits (righ panel) is shown for clarity. The cytoplasmic domain of CpaG and CpaH have similar GspF-like folds each forming a six α-helix motif. They model as an intimate back-to-back heterodimer that spans across two neighbouring CpaF subunits and contacts either Helix 5 conjoined with Loop 8 or the Central Loop. CpaF is modelled with C6 symmetry although based on this study, CpaF likely functions using C2 symmetry in vivo. The left zoom panel highlights the positioning of predicted H0 at the mouth of the nucleotide-binding pocket. The NTD0 of subunit *n* + 1 also contacts H0, which hints at a mechanism for crosstalk between neighbouring subunits. The right zoom panel highlights how two glycine residues within the tip of each CpaG and CpaH α-helix motif are central to the contact zone with the CpaF Loop 8 and Central Loop. These glycines are conserved across type 4 filament family homologues including the T2SS (GspF) and type 4 pilus system (PilC). **B** Analysis of CpaF surface topography predicted to interact with CpaG and CpaH. (Central panel with zoom box) Exploded view of CpaF highlighting the various elevations of Helix 5 conjoined with Loop 8 in the NTD2 relative to the Central Loops in CTD. The Central Loops are observed at high ($CL_H$), medium ($CL_M$) and low ($CL_L$) elevations relating to closed, open' and open active site states. For clarity, flexible N-terminal residues 1–79 have been omitted. (Left panel) Extraction of key CpaF motifs with the predicted binding interface of CpaG and CpaH shadowed as footprints with each spanning Helix 5, Loop 8 and the Central Loop (CL).

in NTD0 sequence were suggested to account for variations in pilus assembly dynamics[26].

Many other motors from type 4 filament members have been crystallised including PilB[27,28,30], PilT[31–33] and PilF[42] from the type 4 pilus system, and FlaI[36] and GspE[37] from archaeal systems. Multiple combinations of active site conformation and symmetries in the presence of different nucleotides were reported showcasing the conformational promiscuity of these motors. However, at least for PilB and PilT, complementary structure determination using cryo-EM has also been undertaken concluding a model where the C2 symmetric hexamer in open/closed/closed states support clockwise motion versus open/closed/open' states for counter-clockwise motion, respectively[33] (Supplementary Fig. 10). For the unidirectional motor PilB, a nucleotide switch is not readily supported as the active sites in the closed states are not sufficiently solvent accessible to enable nucleotide exchange. Therefore, the earliest type 4 filament member motors may have been similar to CpaF with open/closed/open' states that could be geared for bidirectional motion. As previously suggested[33], the

emergence of unidirectional motors like PilB would then be due to recent evolutionary specialisation[16].

CpaF and the platform proteins CpaG/CpaH share close homology with GspE and GspF in the T2SS, and PilB and PilC in the type 4 pilus system, respectively[11]. In these secretion systems they form essential interactions[43–46] with the PilB/PilC complex located in the cytoplasm and inner membrane[47]. Similarly, CpaF is likely located at the base of the TadPS (Fig. 7A) interacting with CpaG and CpaH to spool pilins to or from the pilus through an unknown mechanism. Based on this homology we chose universally conserved residues within Loop 8 and the Central Loop of CpaF to mutate. These were modelled to interact with CpaG and CpaH using Alphafold-Multimer (Fig. 5A). Loop 8 and the Central Loop were likely essential for TadPS function in vivo presumably as the mutations interfere with CpaG and CpaH binding. This data combined with insights from $CpaF_{AMPPNP}$ and $CpaF_{ADP}$ suggest two models for how CpaF energises the TadPS based on either rotary or gyratory movements of CpaG and CpaH. Both models depend on CpaF nucleotide cycling, which drives large-scale

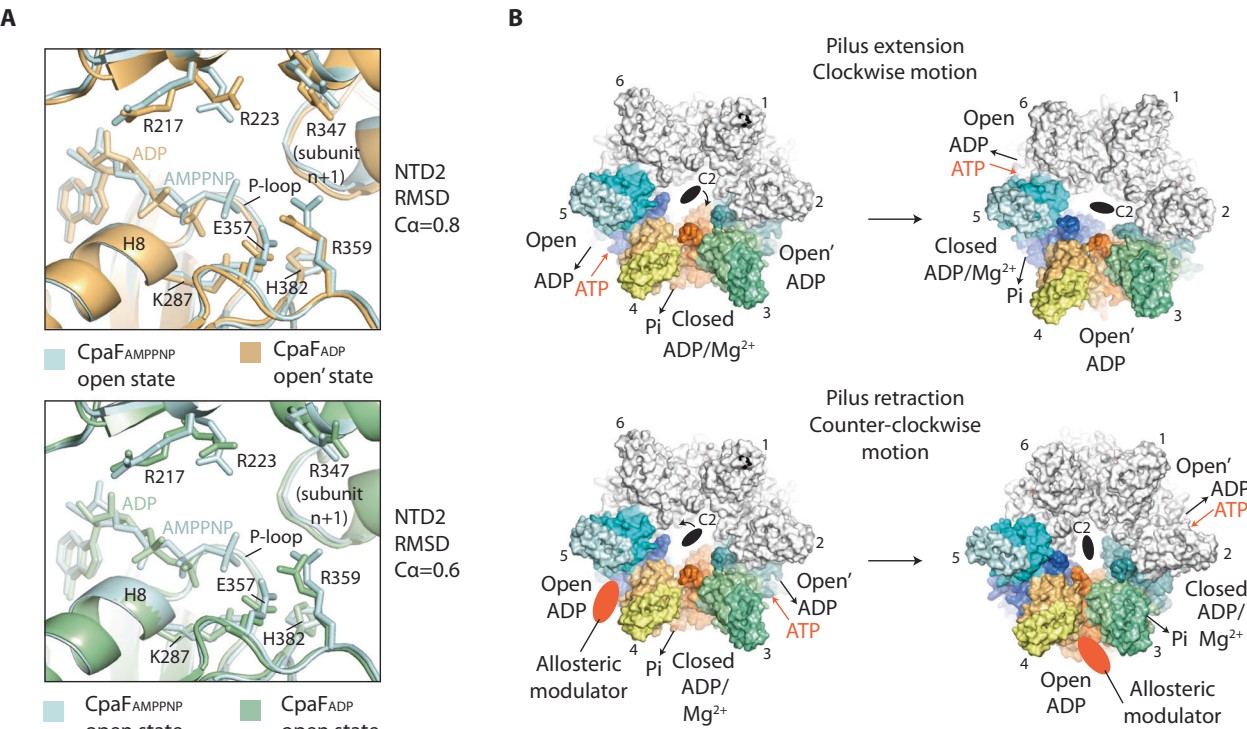

**Fig. 6 | Model for the bidirectional motion of the CpaF motor. A** Superpositions comparing active sites of CpaF$_{AMPPNP}$ open state with CpaF$_{ADP}$ open' and open states. CTDs only were used for each superposition. For clarity, bound Mg$^{2+}$ has been omitted from the CpaF$_{ADP}$ panels. **B** (Top panels) CpaF$_{AMPPNP}$ shows how the open state (AMPPNP bound) has high affinity for ATP, which is then hydrolysed in the closed state (ADP/Mg$^{2+}$ bound) with phosphate release. The open' state has low affinity for ADP and is primed for its release. The sequence of this nucleotide hydrolysis cycle results in clockwise motion of the C2 symmetry axis mediated by large-scale conformational changes between NTD2/CTD$_{n+1}$ units. A clockwise

motion is consistent with the extension of a right-handed pilus. (Bottom panels) Close similarity between the open' state active site in CpaF$_{ADP}$ and the open state in CpaF$_{AMPPNP}$ is consistent with a model where the CpaF open' state has suitable active site geometry to support ATP binding when ADP is bound in the open state. In this way, a counter-clockwise motion would be achieved. An allosteric modulator that may comprise the NTD0 is suggested to act as a gear switch controlling rotation direction possibly by binding and altering the geometry of active sites or nucleotide access and affinity. Other switches are viable such as phosphorylation.

rearrangements in the packing between NTD2/CTD$_{n+1}$ units enabling the C2 axis to rotate around the oval-shaped hexamer (Fig. 6). Concurrently, this motion induces dramatic changes in the surface topography of the hexamer as the Central Loops undergo lever-like conformational changes, shifting through three symmetric elevations spanning ~13 Å (Fig. 5B). A CpaG/CpaH rotary mechanism (Fig. 7B) is best supported if the stoichiometry is one or two CpaG/CpaH heterodimers per CpaF hexamer. As CpaF cycles, the CpaG/CpaH heterodimer sequentially binds to the CpaF subunit pairing for whose surface conformation it has highest affinity. Due to the C2 symmetry, there are two high affinity binding sites per hexamer enabling up to two CpaG/CpaH heterodimers to bind concurrently. This model also predicts that the previously bound site becomes low affinity due to rotation-induced conformational changes. In support of this release mechanism, when the CpaF Central Loop is in the low position it will be blocked from binding CpaG or CpaH due to steric hindrance from the Loop 8 motif above (Fig. 5B). This rotary model differs from that previously suggested for PilB[27] as CpaG/CpaH are not predicted to be positioned in the CpaF central cavity and to be mechanically rotated like a drive shaft. A CpaG/CpaH gyratory mechanism (Fig. 7B) is best supported if the stoichiometry is three CpaG/CpaH heterodimers per CpaF hexamer (Supplementary Fig. 9B) as has recently been shown for PulF/GspF, which models as a C3 symmetric trimer like CpaG/CpaH[48]. In this model, CpaF conformational changes are transmitted directly to the three bound CpaG/CpaH heterodimers which do not rotate relative to CpaF. Instead, the rotation of the CpaF C2 symmetry axis induces a complex gyratory effect in the membrane-spanning domains of the CpaG/CpaH heterodimers, which either move independently of each

other or as a concerted unit. Ultimately, these conformational changes in CpaG and CpaH are somehow coupled to adding or removing pilins from an assembling or retracting pilus.

In conclusion, the comparison of CpaF$_{AMPPNP}$ with CpaF$_{ADP}$ suggests a nucleotide switch model for achieving CpaF bidirectional motion where ATP binding in either the open or open' state stimulates clockwise or counter-clockwise motion, respectively, upon subsequent hydrolysis. However, other mechanisms for how one motor may drive bidirectional rotary motions are feasible with the bidirectional flagellum motor representing one such paradigm. Here, the flagellum is driven by unidirectional rotary motors that deliver both clockwise and counter-clockwise torque simply by engaging different sides of the C-ring rotor in a cog-like mechanism controlled by phosphorylation[49]. Similarly, in the TadPS it cannot be discounted that CpaF has unidirectional motion and other components such as CpaG and CpaH engage different aspects of CpaF structure so as to channel its motion in opposing directions that ultimately drive pilus extension and retraction. This work therefore provides a framework for further studies that characterise the direct interaction of CpaG and CpaH with CpaF and test the models for powering pilus cycling proposed herein.

## Methods
### Bacterial cultures
*Escherichia coli* strain MC1061 was used for cloning and CpaF expression. *E. coli* was grown in LB media with 25 µg/mL chloramphenicol at 37 °C for cloning and propagation of the shuttle vector pINIT. For cloning of the pBXC3H expression constructs and for expression precultures *E. coli* was cultivated in LB media with 100 µg/mL

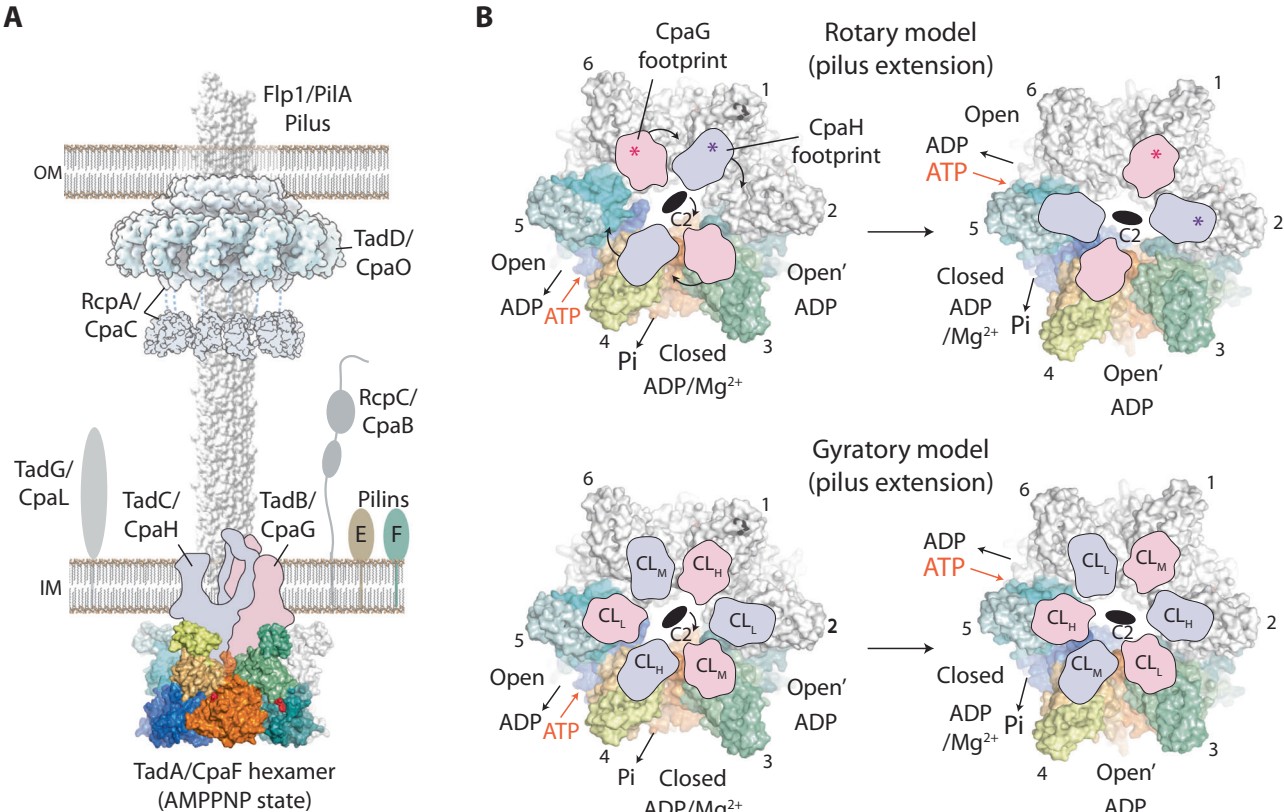

**Fig. 7 | Models for CpaF engagement and energisation of CpaG and CpaH.**
**A** Schematic of the canonical TadPS positioning CpaF in the equivalent position of PilB in the type 4 pilus system[47]. Both canonical TadPS and *C. crescentus* specific Cpa nomenclature is shown. For clarity some canonical TadPS components are omitted including TadV/CpaA and TadZ/CpaE. Single subunits of CpaG and CpaH are depicted as silhouettes based on Alphafold modelling. E and F stand for TadE/CpaJ and TadF/CpaK. Recently it has been suggested that TadG is pilin related and will be positioned at the pilus tip[62]. **B** (Top) Rotary model where a CpaG and CpaH heterodimer, depicted as footprints, has one high-affinity binding interface per CpaF asymmetric unit. As the CpaF C2 symmetry axis rotates due to nucleotide cycling, the CpaG/CpaH heterodimer sequentially binds and unbinds to this

interface resulting in a circular trajectory that is coupled to the spooling of pilins. Given the C2 symmetry, up to two CpaG/CpaH heterodimers could bind and rotate around a single hexamer. **B** (Bottom) Gyratory model based on three CpaG/CpaH heterodimers bound to one CpaF hexamer. This stoichiometry is supported in the T2SS PulF/GspF[48]. As the CpaF C2 symmetry axis rotates, the surface topography at the CpaG/CpaH interface undergoes a complex sequence of conformational changes including the Central Loop moving between high ($CL_H$), medium ($CL_M$) and low ($CL_L$) elevations. The transmission of these conformational changes to CpaG/CpaH would support a gyratory-like movement in the membrane-spanning domains that would be coupled to the spooling of pilins.

ampicillin at 37 °C. *Caulobacter crescentus* NA1000 was cultured in either PYE liquid (2 g/L bactopeptone, 1 g/L yeast extract, 0.3 g/L $MgSO_4$, and 0.0735 g/L $CaCl_2$) or on PYE 1.5% agar plates. Where appropriate, spectinomycin (25 μg/mL) and xylose (0.3%) were supplemented to growth media.

### Construction of *E. coli* expression plasmids
All plasmids used or generated in this study are listed in Supplementary Table 3. *cpaF* (NCBI Gene ID: 7333491) was PCR amplified from *C. crescentus* NA1000 genomic DNA using primers CpaF_for and CpaF_rev (Supplementary Table 2). The purified PCR product was cloned into the shuttle vector pINIT_cat using FX cloning[50] resulting in pINIT_*cpaF*. After verifying the sequence, pINIT_*cpaF* was subcloned into pBXC3H fusing CpaF C-terminally to a 3 C protease cleavage site and a $His_{10}$-tag resulting in the plasmid pBXC3H_*cpaF*. CpaF mutants for in vitro purification were constructed in pBXC3H using FX and Gibson[51] cloning with the primers listed in Supplementary Table 2.

### Construction of *C. crescentus* knock-out and knock-in vectors
Primers NTToligo4020 and NTToligo4021 were used to amplify a 500 bp region upstream of *cpaF* including the first 180 nucleotides of *cpaF*, while primers NTToligo4022 and NTToligo4023 amplified a 500 bp region downstream of *cpaF* including the last 180 nucleotides of

*cpaF*. Primers NTToligo2978 and NTToligo2979 were used to amplify a 500 bp region upstream of *pilA* including the first 9 nucleotides of *pilA*, while primers NTToligo2980 and NTToligo2981 amplified a 500 bp region downstream of *pilA* including the last 9 nucleotides of *pilA*. Primers NTToligo2978 and NTToligo4018 were used to amplify a 500 bp region upstream of *pilA* and the first half of *pilA* with the T36C mutation, while primers NTToligo4019 and NTToligo2981 amplified a 500 bp region downstream of *pilA* and the second half of *pilA*. All PCR products were cloned into suicide vector pNPTS138 using Gibson assembly, resulting in the knock-out plasmids pNPTS138::Δ*cpaF* and pNPTS138::Δ*pilA*, and the knock-in plasmid pNPTS138::*pilA* (T36C). All constructs were verified by Sanger sequencing.

### Construction of *C. crescentus* complementation plasmids
Double-stranded DNA fragments containing *cpaF* wildtype and mutant genes with a 3' FLAG tag were chemically synthesised as gBlocks (IDT). These gene fragments were cloned into vector pXYFPC-1[52] after removing the *yfp* gene using Gibson assembly[51]. Briefly, 2.5 μL of each gBlock fragment and 2.5 μL of NdeI-NheI-cut pXYFPC-1 were added to 5 μL Gibson master mix (NEB) and incubated at 50 °C for 60 minutes. Five μL were used to transform chemically competent *E. coli* DH5α cells. The resulting pXYFPC-1::*cpaF* plasmids contained either the wildtype *cpaF* or mutant genes and were verified by Sanger sequencing.

## Construction of *C. crescentus* strains

All strains generated in this study are listed in Supplementary Table 3. The suicide plasmids pNPTS138::Δ*cpaF*, pNPTS138::Δ*pilA* and pNPTS138::*pilA* (T36C) were electroporated into *C. crescentus* NA1000 cells. Double-crossover exconjugants were generated by sucrose counter-selection as described previously[53] resulting in the *C. crescentus* strains NA1000 Δ*cpaF*, NA1000 Δ*pilA* and NA1000 *pilA* (T36C). Markerless gene deletion mutants were identified by PCR. To construct Δ*cpaF* complementation strains, electrocompetent *C. crescentus* NA1000 *pilA* (T36C) Δ*cpaF* cells were transformed individually with complementation plasmids pXYFPC-1::*cpaF* containing either the wildtype *cpaF* or mutant genes to allow for a single integration at the *xylX* locus. The correct integration was verified by PCR.

## PhiCbK phage-sensitivity assays

*C. crescentus* strains were inoculated into 10 mL of PYE and cultured for 20 hours at 30 ˚C with orbital shaking at 250 rpm. Cultures were diluted and grown to an $OD_{600} = 0.2$–0.3. Subsequently, strains were adjusted to an $OD_{600} = 0.2$ and 100 μL were mixed with 10 mL of soft PYE 0.5% agar and overlaid onto a solid PYE 1.5% agar base. Five μL of ten-fold serial dilutions of phiCbK phage stock were spotted in triplicate onto each plate. Finally, plates were incubated at 30 ˚C overnight and imaged the next day. Plaque formation indicated sensitivity to phiCbK phage.

## Pili labelling and fluorescence microscopy

*C. crescentus* strains were inoculated into 10 mL PYE and cultured for 20 h at 30 ˚C with orbital shaking at 250 rpm. Cultures were then diluted and grown to an $OD_{600} = 0.2$–0.3. Alexa Fluor 488 C5 Maleimide (Invitrogen) was added to 100 μL of *C. crescentus* culture (final concentration: 25 μg/mL) and incubated at room temperature for 10 minutes. Samples were centrifuged at $2500 \times g$ for one minute and the pellet was carefully resuspended in 100 μL PBS. Cultures were centrifuged again, and the pellet was resuspended in a final volume of 50 μL PBS. Cells were immobilised on 1 % PYE agarose pads and imaged using a Zeiss Axio Observer Z.1 inverted epifluorescence microscope equipped with a sCMOS camera (Hamamatsu Orca FLASH 4) and Zeiss Colibri 7 LED light source. Images were captured with a Zeiss Plan Apochromat 100x/NA 1.4 Ph3 objective lens and a GFP filter set (excitation: 488 nm, emission: 500–549 nm) using an exposure time of 250 ms. Images were acquired using Zen Blue 2.5 (Zeiss) software and subsequently analysed using ImageJ 1.54[54]. The MicrobeJ plug-in for Fiji[55] was used to detect *C. crescentus* cells and to measure the background-corrected mean fluorescence intensity within individual cells. The maximum fluorescence intensity for the non-fluorescent wildtype *C. crescentus* established an autofluorescence baseline; any value above this was categorised as true fluorescence. Each cell was also manually inspected to determine presence or absence of pili and used to calculate the proportion of the cell population that possessed extended pili.

## Western blots

*C. crescentus* strains were inoculated into 10 mL of PYE and cultured for 20 hours at 30 ˚C with orbital shaking at 250 rpm. Cultures were diluted and grown to $OD_{600} = 0.2$–0.3 and 5 mL were centrifuged at $1450 \times g$ for 10 minutes. Cell pellets were resuspended in 300 μL lysis buffer (20 mM HEPES pH 7.9, 50 mM KCl, 10% glycerol, protease inhibitor tablet) and sonicated on ice. The cell lysate was centrifuged at $17,000 \times g$ for 15 minutes at 4 ˚C. Samples of the supernatant were boiled for five minutes in SDS-PAGE loading dye (containing β-mercaptoethanol) before separation by SDS-PAGE together with a coloured protein broad range standard (NEB). Proteins were transferred onto a PVDF membrane using a Trans-Blot Transfer System (BioRad) and the membrane was blocked in TBS, 0.1% Tween-20, 5% milk powder for 1 h. The membrane was then incubated with a 1:5000

dilution of a monoclonal α-FLAG HRP-conjugated antibody (Sigma, A8592) for 1 h. Finally, the membrane was washed five times in TBS, 0.1% Tween-20 for one minute each and then briefly incubated with SuperSignal™ West Femto Maximum Sensitivity Substrate (Thermo Scientific™). Blots were visualised using an Amersham Imager 600 (GE Healthcare).

## Expression and purification of *C. crescentus* wildtype CpaF

CpaF was purified from *E. coli* MC1061. Glycerol stocks of *E. coli* MC1061 harbouring the plasmid pBXC3H_*cpaF* were used to inoculate a LB broth preculture grown at 37 ˚C overnight. The preculture was diluted 1:40 (v/v) into 2xYT and grown at 37 ˚C for two hours and subsequently at 25 ˚C for one hour to an $OD_{600}$ of -0.8–1.0 before induction of protein expression with 0.01% L-arabinose. The next day cells were harvested at $9000 \times g$ for 20 minutes at 4 ˚C. Pellets were resuspended in cold lysis buffer containing 20 mM Tris/HCl pH 8, 200 mM NaCl, 3 mM MgSO$_4$, DNase I (Sigma) and one tablet of complete EDTA-free protease inhibitor cocktail (Roche). The cells were disrupted by three passages through an Avestin EmulsiFlex-C5 homogeniser at 20–25 kpsi keeping the samples on ice. Unbroken cells were removed by centrifugation at $27,000 \times g$ for 30 minutes at 4 °C. The supernatant was loaded on Ni$^{2+}$-NTA (Qiagen) columns. The column was washed with 200 mM imidazole pH 7.5, 200 mM NaCl, 10% glycerol and 1 mM PMSF. CpaF was eluted with 500 mM imidazole pH 7.5, 200 mM NaCl, 10% glycerol and 1 mM PMSF. To remove the C-terminal His$_{10}$-tag the elution fraction was incubated with 3 C protease overnight at 4 ˚C. Finally, CpaF was separated from the cleaved His$_{10}$-tag and the 3 C protease via size exclusion chromatography using a HiPrep 26/60 Sephacryl S-300 HR column equilibrated in 20 mM HEPES pH7.4, 150 mM NaCl, 3 mM MgSO$_4$.

## Negative stain microscopy of CpaF mutants

CpaF mutants were purified like wildtype except a Superose 6 10/300 GL column was used for the final gel filtration. For negative stain EM, 3.5 μL of protein sample was applied to glow discharged carbon coated 300 mesh copper grids (Agar scientific). Grids were washed with three drops of water and then stained with 2 % uranyl acetate. Images were acquired on a FEI Tecnai TEM 120 keV microscope equipped with a FEI 2 K Eagle camera.

## Cryo-EM grid preparation and CpaF data collection

5 mg/mL CpaF was used for vitrification with 8 mM CHAPSO added to all samples to mitigate particle preferential orientation. For nucleotide associated samples, 4 mM ADP or AMPPNP (Sigma) was also added with a total of 5 mM MgSO$_4$ and a 45 minute incubation at room temperature. Samples were kept on ice until freezing in liquid ethane using glow-discharged 300 mesh holey carbon copper grids (Quantifoil R2/2, Electron Microscopy Sciences). The CpaF-APO dataset was acquired on a 300 kV Titan Krios electron microscope (LonCEM, The Francis Crick Institute, London, UK) equipped with a Gatan K3 detector operated in super-resolution mode with a pixel size of 0.425 Å. 21,498 movies were collected using a total dose of 40 e⁻/Å² fractionated over 44 frames with 2.3 second exposure time and defocus range of −0.5 to −2.5 μm. The CpaF-ADP dataset was collected on the 300 kV Titan Krios IV microscope at eBIC (Diamond Light Source, Didcot, UK) equipped with a Gatan K3 detector operating in super-resolution mode with a pixel size of 0.4145 Å. 24,809 movies were collected using a total dose of 40 e⁻/Å² fractionated over 40 frames with 1.8 second exposure time and defocus range of −0.5 to −2.1 μm. The CpaF-AMPPNP dataset collection used the 300 kV Titan Krios at LonCEM. The dataset was acquired in super-resolution mode with a pixel size of 0.55 Å. 5108 movies were collected using a total dose of 40 e⁻/Å² fractionated over 40 frames with 4.39 second exposure time and defocus range of −0.5 to −2.8 μm.

## CpaF cryo-EM data processing

The APO dataset was processed using cryoSPARC v4.0.1[56] and later using Relion 3.1[57]. Micrographs were aligned using patch motion correction and Fourier cropped to 0.85 Å/pixel. CTF patch estimation was performed, and 19,493 of the best micrographs were selected for further processing. Template picker yielded 2,920,875 particles which was reduced to 471,974 particles after five rounds of 2D classification. The stack was transferred to Relion 3.1 for further 2D classification resulting in 440,053 particles. An initial reference was generated in Relion 3.1 using a stochastic descent gradient algorithm. 3D classification with C1 symmetry was undertaken to determine symmetry. C2 symmetry was subsequently applied to all processing steps. After several rounds of 3D classification, a final stack comprising 51,697 particles was used for iterative rounds of 3D autorefinement incorporating two rounds of CTF refinement and Bayesian polishing. The final resolution of 4.0 Å was based on gold standard Fourier shell correlations (FSC) = 0.143 criterion. For the ADP dataset, MotionCor2 was used to align movies and Fourier crop to 0.829 Å/pixel. CTFFIND-4.2 was used for CTF estimation. Using Relion 3.1, 23,774 micrographs were selected for particle picking resulting in 4,630,409 particles. After four rounds of 2D classification, 560,951 particles were used for iterative rounds of 3D classification. Using a final stack of 114,641 particles, a similar refinement strategy as for CpaF-APO was followed yielding a map at 3.1 Å resolution overall using gold standard FSC = 0.143 criterion. The CpaF-AMPPNP dataset was processed equivalently to CpaF-ADP. MotionCor2 was used to align movies and Fourier crop to 1.1 Å/pixel. CTFFIND-4.2 was used for CTF estimation. Using Relion 3.1, 5,013 micrographs were selected for particle picking resulting in 419,228 particles. After six rounds of 2D classification, 204,843 particles were used for iterative rounds of 3D classification. Using a final stack of 73,200 particles, a similar refinement strategy as for CpaF-APO was followed yielding a map at 3.8 Å resolution overall using gold standard FSC = 0.143 criterion. Map local resolution was estimated using ResMap[58]. CpaF-APO, CpaF-AMPPNP and CpaF-ADP maps were sharpened using Phenix 1.2[59].

## Model building and refinement

CpaF models were built in Coot[60] for each asymmetric unit from amino acids 150–502. No supporting map was observed for residues 1–79, which were predicted to be disordered. Map relating to the NTD1 (residues 80–149) was poorly resolved due to flexibility in Hinge 1. Particle subtraction masking out a single protomer of the AMPPNP map followed by focussed refinement improved the map quality in the NTD1 so that a three-helix motif could be readily traced with side chain detail limited to bulky residues. An NTD1 model was generated by Alphafold[34], a rigid body fitted and modelled using ISOLDE[61]. This NTD1 model was then used for rigid body fitting in all other NTD1 map regions and joined with the NTD2/CTD chains. Models for the complete asymmetric unit were real space refined using Phenix 1.2[59] but with NTD1s defined as rigid bodies. Finally, a C2 operator was used in Phenix[59] to reconstitute the CpaF hexamer.

## Alphafold-Multimer modelling

CpaF, CpaG and CpaH in 6:3:3 stoichiometry were modelled with Alphafold v2.3.2 installed locally using default 'multimer' parameters[41]. Due to processing limitations, one model was generated with an overall per-residue local distance difference test (pLDDT) = 74.9. The pLDDT scores for individual subunits were retrieved from the B-factor column of the pdb file (Supplementary Fig. 9C). Individual pLDDT scores for CpaF, CpaG and CpaH were 90.0, 80.1 and 82.0, respectively.

## Reporting summary

Further information on research design is available in the Nature Portfolio Reporting Summary linked to this article.

## Data availability

The cryo-EM maps generated in this study have been deposited in the Electron Microscopy Data Bank under accession codes EMD-19275 (CpaF-AMPPNP), EMD-19209 (CpaF-ADP) and EMD-19279 (CpaF-APO). The atomic coordinates have been deposited in the Protein Data Bank (PDB) under accession codes 8RKD (CpaF$_{AMPPNP}$), 8RJF (CpaF$_{ADP}$) and 8RKL (CpaF$_{APO}$). Source Data are provided as a Source Data file. Source data are provided in this paper.

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

## Acknowledgements

We thank Nora Cronin (LonCEM, The Francis Crick Institute) and Karen Davies (eBIC, Diamond Light Source) for cryo-EM data collection support and Paul Simpson for in-house EM support. We thank Ngat T Tran for some strain constructions for preliminary experiments. This work was funded by a Wellcome Trust Senior Research Fellowship (215553/Z/19/Z)

to HL. This work was also supported by a Royal Commission for the Exhibition of 1851 Research Fellowship (E.J.B), a Lister Institute Prise Fellowship (T.B.K.L), the Wellcome Trust Investigator Grant 221776/Z/20/Z (T.B.K.L), and the BBSRC-funded Institute Strategic Programme Harnessing Biosynthesis for Sustainable Food and Health (BB/X01097X/1).

## Author contributions

All authors designed experiments. MH purified proteins and undertook negative stain and cryo-EM studies, including sample preparation, data collection, processing, and model building. HL contributed to data collection and processing. MM contributed to data processing. EJB undertook *C. crescentus* cell-based assays including strain generation, phage infection assays and fluorescent light microscopy. HL and TL supervised the work. HL wrote the paper with contributions from all authors.

## Competing interests

The authors declare no competing interests.
