## [Peer Review File · Nature Communications]

Bidirectional pilus processing in the Tad pilus system motor
CpaFREVIEWER COMMENTS

Reviewer #1 (Remarks to the Author):

In this manuscript, Hohl and co-workers report the cryo-EM structure of the TAD pilus ATPase, TadA. Critically, they have obtained the structure of this complex in various nucleotide states, providing the molecular basis for its dual role in plus extension and retraction. Finally, they use a clever assay for structure-function characterisation of this complex, supporting a direct interaction with the TAD plus IM complex.

This is a very neat study, that provides important new insights into the structure and mechanism of assembly of the TAD pilus, and of the T4P family in general. While there are no major surprises, the experiments are performed carefully, and the conclusions are extremely well supported by the reported data. I am therefore supportive of publishing this work in Nature Communications.

I only have a few minor comments:

- I am not convinced that the term "Tad Secretion System", TadSS, is really suitable: There is only one secreted protein, Flp, so it does not really constitute a secretion system; and this is not a term that is commonly used in the T4P field either. I am aware that the authors employed this term in their previous publication on the Tad secretin, so there is precedent, but I would prefer to keep the standard terminology used in the field, i.e. Tad pilus (with reference to its assembly machinery as required).
- Page 3: The authors should clarify that the ATPase has been shown to be bi-directional in *C. crescentus* only, it is unclear if retraction occurs in other bacteria.
- First result section, and Figure 1: The authors report the presence of ADP in two of the nucleotide-binding pockets. Where do the authors think this ADP comes from, and how can they tell if is not simply poor density for an AMPPNP molecule? If it was dragged through the purification process, why do they not see any nucleotide in their apo structure? This needs to be further clarified.
- In the modelling of TadABC (Figure 5), the pLDDT plot should be added to the supplementary data. Have the authors tried several stoichiometries for TadB and TadC?
- I am not convinced that Figure 6 is really necessary, as all the information shown there is already contained in figure 7B. However, an additional figure comparing the mechanism of TadA to that of PilB and PilT would greatly facilitate reading of the discussion.

Reviewer #2 (Remarks to the Author):

Comments to the authors

In their manuscript, the authors address the unknown mechanism of the ATPase TadA, which

energizes both extension and retraction of the Tad pilus in many different bacteria. The focus lies on structural analysis (cryo-EM and modelling), which is complemented by cell-based assays to determine the effect of selected mutations in vivo.

The paper is very well written, and most data and results are clearly explained. The findings provide new and interesting information, and should appeal to a broad readership given the wide distribution of bacterial Tad systems.

My comments are:

1. page 6/7/8 and respective figures: The structures and dynamics of specific positions are described in great detail. However, not all of the descriptions are readily represented in the figures. To improve readability, I ask the authors to re-check (i) where additional features should be highlighted in the figures, (ii) which highlighted features might not be discussed and could be omitted from the figures, or (iii) where perhaps references to other figure should be included. Examples are: loop 26 in Fig. 1A (not in text), CTD loop (not in figure), sheet 12 & 13 (not in figure), Walker B E357 (not Fig. 2C but in Fig. 4A).

2. I don't fully understand the manuscript's contribution to the TraA pilus retraction function. In the fluorescence microscopy assay, a retraction-specific phenotype (page 11, line 4) was not observed. Did the authors expect to observe such a phenotype from the selected mutants, and what does it mean that they didn't? How could their proposed mechanistic model for bidirectional rotation be experimentally verified? Also, it is confusing that the text in the discussion repeatedly mentions "reverse rotation" while corresponding Fig 6 states "counter clockwise rotation". And how does that relate to the nucleotide switch?

I ask the authors to please comment on this. Also, re-wording parts of the discussion to better explain the main conclusion and also its limitations are necessary to make this paper more accessible for the non-specialist reader.

Reviewer #3 (Remarks to the Author):

In this manuscript, the authors use CryoEM to resolve the structure of TadA, the ATPase driving the extension and retraction of Type IV-like Tad pili in *Caulobacter crescentus*. Hexameric structures of the TadA apo form and of TadA in the presence of AMP or AMPPNP, a non-hydrolysable analog of ATP, show a characteristic symmetric arrangement of alternate open and closed monomer conformations, similar to what was reported for related bacterial pili ATPases and ATPases of the pili-like archaeal motor. These studies reveal atomic details of nucleotide binding to TadA, based on which the authors postulate an ATP-dependent power stroke model in line with previous reports, where sequential ATP binding, catalysis and release drive TadA rotation. Finally, alphafold predictions are used to define interaction surfaces between TadA and the pili platform proteins TadB and TadC. Mutational analyses coupled to functional assays confirm that conserved residues of proposed interaction surfaces are required for Tad pili function.

Overall, the authors' work nicely resolves the structure of TadA and its nucleotide binding conformations with reasonable resolution and convincing ligand binding information. This is the first structure of an ATPase of the T4cP subfamily, which was recently proposed to be able to rotate in both directions, representing an evolutionary ancestor of the widespread unidirectional pili ATPases. These results provide the basis for a better understanding of ATPase reversibility and the authors put forward an interesting model for bi-directional rotation. However, the experimental findings substantiating bidirectional activity of TadA remain premature and the model entirely speculative.

Major points:

- The mechanistic model of TadA clockwise rotation for pilus extension and counterclockwise rotation for pilus retraction is based on the similarities between the P-loop disorder of the AMPPNP and ADP structures (page 13, 1st paragraph). A map of this region (P-loop, active site) of the two different structures would help readers to be able to better understand the proposed model.

- TadA mutations impacting ATPase activity have been described previously (PMID: 31897429). The observation that these mutations equally affect pili extension and retraction, has led to the hypothesis of TadA bidirectionality. Can the authors rationalize these phenotypes based on their structural data? Likewise, can the structure predict residues that are essential for bidirectionality? One would expect that such residues are conserved among bidirectional orthologs and missing in their uni-directional counterparts. If such residues can be identified, is it possible to generate unidirectional (extension only) TadA mutants?

- The role of Mg is unclear and should be discussed by the authors. In the methods it is stated that lysis was done in a buffer containing magnesium, but it is unclear if Mg was added after this first step and could have been washed away during purification. Could this explain why Mg is found only in the closed state of TadA but not in the Open' state in the presence of AMPPNP? And why is Mg only found in the Open' state when AMPPNP is absent? Can the authors exclude the possibility that by saturating the amounts of Mg, the ADP open' would also contain a Mg ion and that this could change the conformational state (e.g., to closed state)? What is the distance of Mg to ADP and to E357 (not shown in Figs. 2 and 3)? Does any other residue participate in the coordination? How is E357 oriented in the other subunits of TadA-AMPPNP without Mg in comparison to the Mg bound subunits?

- The observation that TadA-AMPPNP, TadA-ADP and TadA apo structures are very similar, suggests that the structural states are not driven by bound nucleotides, but rather are an inherent feature of the protein hexamer under the conditions used for CryoEM. What is the experimental evidence that open and closed states are indeed determined by the bound nucleotides?

- The details of the alphafold modeling of the interaction between TadA and the platform proteins are not provided by the authors and should be included in the Methods section. It looks like 2x TadA, 1x TadB and 1x TadC was modelled. Does AF give the same interface of the two TadA subunits as the cryo-EM structure or are they different? Do TadB and TadC interact? Are there major clashes

in the model? Also, most TadA mutants show reduced levels and potential degradation products (Supplementary Figure 7A) indicate that these constructs may not be physiological. Are the observed phenotypes indeed based on disrupted protein-protein interaction or are they the result of limited overall protein concentration/stability?

- On page 14 and Figure 7, the authors discuss the stoichiometry differences of their models. However, in Figure 5, only one TadB and one TadC protomer are bound to a TadA hexamer. Can the authors model 2 or 3 TadB/C pairs to one TadA hexamer, to investigate potential collisions? Would TadA residues relevant for TadB/C interaction remain at the interface?

- Structural details in figures 2 and 3 are hard to grasp and color codes are confusing.

- I would recommend switching the order of Figs. 4 and 5 and change the text on p.10/11 accordingly. It is difficult for the reader to understand the functional assays without the interaction model having been introduced.

Minor points:

- The type IV pili field is notorious for its lack of a common nomenclature for conserved components of the same machinery in different organisms. However, one would expect that the same nomenclature is used at least for the same species. Other reports on the *Caulobacter* pilus system have used a different nomenclature (CpaF for TadA and PilA for Flp1) and, to avoid confusion, I would strongly advise to adhere to this terminology.

- In Supplementary Figure 7C, sample 4 shows lower distribution compared to wild type (sample 1 and 3). This should be mentioned and discussed in the main text as evidence for the functionality of N-term region. It is also interesting that there is no difference in the frequency of piliated cells (Figure 7D). Does this suggest that the N-term region is important for effective retraction, but not is affecting the extension of the pili (thus leading to less cell-body label from extra-cellularly labelled pili)? The phage assays will not contradict this result, as phages may be able to infect even the retraction is less effective. Please comment.

- Page 12: “confirming a loss of pilus extension rather than just a retraction defect”. It would be easier for the reader to understand this part if the observed retraction defects of other pili systems are mentioned, such as elongated pili in retraction ATPase mutant in other organisms.

We would like to thank the reviewers very much indeed for taking the time to critique our manuscript so thoroughly and constructively. We really appreciate it. The manuscript has benefited from your comments.

Note we have updated the nomenclature used in this reviewer response to match that used in the manuscript.

REVIEWER COMMENTS

Reviewer #1 (Remarks to the Author):

In this manuscript, Hohl and co-workers report the cryo-EM structure of the TAD pilus ATPase, CpaF. Critically, they have obtained the structure of this complex in various nucleotide states, providing the molecular basis for its dual role in plus extension and retraction. Finally, they use a clever assay for structure-function characterisation of this complex, supporting a direct interaction with the TAD plus IM complex.

This is a very neat study, that provides important new insights into the structure and mechanism of assembly of the TAD pilus, and of the T4P family in general. While there are no major surprises, the experiments are performed carefully, and the conclusions are extremely well supported by the reported data. I am therefore supportive of publishing this work in Nature Communications.

We thank the reviewer for their positive comments and support to publish in Nature Communications.

I only have a few minor comments:

1) I am not convinced that the term "Tad Secretion System", TadSS, is really suitable: There is only one secreted protein, Flp, so it does not really constitute a secretion system; and this is not a term that is commonly used in the T4P field either. I am aware that the authors employed this term in their previous publication on the Tad secretin, so there is precedent, but I would prefer to keep the standard terminology used in the field, i.e. Tad pilus (with reference to its assembly machinery as required).

Tad secretion system was used in our recent Nature Comms 2023 paper as we were following the original terminology used by the Figurski lab who were involved in the discovery of the TadSS¹ and really set the scene for the field over a decade circa 2000-2010. Tad secretion system was their routine nomenclature²⁻⁵ as exemplified in their 2007 review abstract "Here we review the structure, function and evolution of the Tad secretion system". By this time the Figurski lab were likely aware that the Flp pilin was the prime substrate and continued to describe the assembly machinery as a secretion system possibly because *Aggregatibacter actinomycetemcomitans* secretes copious pili as biofilms. Given this we don't think we were incorrect to use the TadSS nomenclature in our prior 2023 paper. However, in a bid to align the field we are happy to use Tad pilus system (TadPS) in this study and have made the necessary changes to the manuscript.

2) Page 3: The authors should clarify that the ATPase has been shown to be bi-directional in *C. crescentus* only, it is unclear if retraction occurs in other bacteria.

Good point. We have modified the text at L106-109 to read: In the *C. crescentus* TadPS, pilus extension and retraction are mediated by the ATPase CpaF alone⁶ – yet the mechanism for how one motor drives bidirectional pilus cycling remains elusive. It is also unknown whether this motor bifunctionality is conserved across the TadPS family or is unique to *C. crescentus*.

3) First result section, and Figure 1: The authors report the presence of ADP in two of the nucleotide-binding pockets. Where do the authors think this ADP comes from, and how can they tell if it is not simply poor density for an AMPPNP molecule? If it was dragged through the purification process, why do they not see any nucleotide in their apo structure? This needs to be further clarified.

The CpaF_{apo} structure confirms that without the addition of extraneous nucleotide, no nucleotide is bound during purification and all nucleotide binding pockets are empty in the cryo-EM map. Given this, we conclude that the ADP observed in the CpaF-AMPPNP map results from AMPPNP hydrolysis. This is not surprising as non-hydrolysable ATP analogues are not entirely resistant to hydrolysis⁷. Crucially, map quality is unambiguous in the closed state nucleotide binding pocket where ADP and Mg²⁺ are bound – AMPPNP is not bound here. The neighbouring pocket where ADP is fitted has a disordered P-loop and generally the binding pocket is less structured as β -sheets 12 and 13 are partially disordered. Here, the map supports the fitting of loosely bound ADP primed to exit so that a new hydrolysis cycle can initiate.

To clarify to the reader why ADP ligands are observed in the CpaF_{AMPPNP} map/model we now state in L164-169: Although CpaF was incubated with AMPPNP, this nucleotide was symmetrically bound in only two of the nucleotide binding pockets within the hexamer. ADP coordinated to a Mg²⁺ ion (ADP/Mg²⁺) was symmetrically bound in the neighbouring subunit, with ADP alone in the remaining pair (Figure 1A). Here ADP is likely derived from AMPPNP hydrolysis given non-hydrolysable ATP analogues are not entirely resistant to hydrolysis⁷.

4) In the modelling of CpaFBC (Figure 5), the pLDDT plot should be added to the supplementary data.

The pLDDT plots are now shown in Supplementary Figure 9C. Please also see our response to #23 below.

5) Have the authors tried several stoichiometries for CpaG and CpaH?

Reviewer 3 asks a similar question but with expanded detail. Please see #23 below.

6) I am not convinced that Figure 6 is really necessary, as all the information shown there is already contained in figure 7B.

We understand the reviewer's concerns and have considered merging Figure 6 with Figure 7B as suggested. However, we conclude that this may detract from the clarity of the figures and paper and possibly confuse readers. We think it is important that Figure 6 stands as a single message to describe our model for how CpaF mediates bidirectional motion. The figure contains important details such as the allosteric modulator that is not immediately relevant for Figure 7. Additionally, the figure now includes Figure 6A to highlight the similarity between active sites of CpaF_{AMPPNP} open state with CpaF_{ADP} open' and open states. Similarly, for Figure 7 we think it important that the focus is solely on how CpaF engages CpaG and CpaH, which is independent of the CpaF bidirectionality model. As it stands Figure 7 does not show how CpaF may promote counter-clockwise motion and incorporating it would make the figure complicated. Given this we would like to keep Figures 6 and 7 separate please.

7) However, an additional figure comparing the mechanism of CpaF to that of PilB and PilT would greatly facilitate reading of the discussion.

In support of this welcome suggestion, we have generated Supplementary Figure 10 referenced in the Discussion. This showcases the mechanism of PilB and PilT which the reader can view with comparable figures of CpaF e.g. Figure 6. We did not include CpaF directly in the supplementary figure to avoid duplication with Figure 6. We have written extensive commentary in the figure legend describing highlights of PilB and PilT mechanism⁸⁻¹⁰. Note also that the molecular detail of PilT catalytic mechanism is still elusive as many crystal structures show PilT in multiple conformations and nucleotide states. The PilT cryo-EM structure used to conclude that the *in vivo* form was likely C2 symmetry was nucleotide-free and relatively low resolution (4.1 Å; PDB 6OLL). We have therefore avoided discussing the mechanism beyond the overall conformation/state of the subunits.

Reviewer #2 (Remarks to the Author):

Comments to the authors

In their manuscript, the authors address the unknown mechanism of the ATPase CpaF, which energizes both extension and retraction of the Tad pilus in many different bacteria. The focus lies on structural analysis (cryo-EM and modelling), which is complemented by cell-based assays to determine the effect of selected mutations *in vivo*. The paper is very well written, and most data and results are clearly explained. The findings provide new and interesting information, and should appeal to a broad readership given the wide distribution of bacterial Tad systems.

We thank the reviewer for their positive comments and highlighting how our manuscript should appeal to a broad readership.

My comments are:

8) page 6/7/8 and respective figures: The structures and dynamics of specific positions are described in great detail. However, not all of the descriptions are readily represented in the figures. To improve readability, I ask the authors to re-check (i) where additional features should be highlighted in the figures, (ii) which highlighted features might not be discussed and

could be omitted from the figures, or (iii) where perhaps references to other figure should be included. Examples are: loop 26 in Fig. 1A (not in text), CTD loop (not in figure), sheet 12 & 13 (not in figure), Walker B E357 (not Fig. 2C but in Fig. 4A).

Thank you for catching these discrepancies. We have rechecked everything and made the following amendments/corrections:

-Loop 26 is now included in the text replacing CTD loop: 'Consequently, the CTD/CTD_{n+1} cleft is compressed with no HEPES bound, and Loop 26 between β -sheets 14 and 15 is now directly contacting H13 in CTD_{n+1} (Figure 2A).

-S13 relabelled correctly as S14 in Figure 1B.

-S13 and S14 relabelled correctly as S14 and S15 in Figure 2A.

-E357 has now been included in Figure 2C and Figure 3C. It was previously omitted so as to not overly complicate the figure but as part of the Walker B motif we acknowledge it needs to be there.

-E312 (ASP box) has been included in Supplementary Figure 5. It has not been added to Figure 2C and Figure 3C as its position clashes with other residues and it will reduce clarity. Its omission has been described in the legend and the reader directed to Supplementary Figure 5.

9) I don't fully understand the manuscript's contribution to the CpaF pilus retraction function. In the fluorescence microscopy assay, a retraction-specific phenotype (page 11, line 4) was not observed. Did the authors expect to observe such a phenotype from the selected mutants, and what does it mean that they didn't?

Supplementary Figure 7C (now Figure 4E right panel) showed a significant reduction in the proportion of fluorescent cells ($p=0.0022$) for the $\Delta 1-78aa$ strain compared with the Flp1 T36C strain (as Reviewer 3 highlights #30). This indicates the NTDO functions as a positive regulator of pilus retraction. Whilst we were aware of this finding, we did not previously highlight it in the text as i) we were overly cautious at the time to not over-interpret, ii) we had not spotted corroborating work from the Brun lab where they show that the NTDO sequence likely accounts for variations in pilus assembly dynamics between *C. crescentus* and *Asticcacaulis biprosthicum*⁶. As this data suggests the NTDO to be an important regulator of pilus dynamics and motor bifunctionality, we have now:

i) moved Supplementary Figure 7C and 7D into the main text as Figure 4E.

ii) discussed in the Discussion L413-424 how the NTDO may act allosterically to regulate pilus retraction and possibly motor directionality.

iii) included in Figure 5A a zoom panel that shows the AlphaFold modelled position of NTDO Helix 0 locating to the mouth of the nucleotide binding pocket. The implications of this are included in the discussion above.

10) How could their proposed mechanistic model for bidirectional rotation be experimentally verified?

We are keen to test our model although this is outside the scope of this current study. As part of a future 2-3 year workplan and funding cycle we will attempt: i) to determine nucleotide

ligand affinity and relative stoichiometry for CpaF using isothermal titration calorimetry (ITC); ii) to dissect how the NTDO modulates pilus dynamics using FRET-like photoinduced electron transfer (PET) assays; iii) to investigate TadPS pilus dynamics in other bacteria with variable NTDO lengths and to assess NTDO chimeras between CpaF homologues; iv) to undertake molecular dynamic simulations using all-atom free-energy simulations to capture clockwise and counter-clockwise motion. Promising outcomes from these methods will also provide the foundation of future cryo-EM studies to obtain counter-clockwise motion-supporting CpaF states.

Note that as part of this revision we undertook ATPase assays to probe whether the NTDO has the potential to modulate CpaF activity. The NTDO does not affect nucleotide turnover *in vitro*.

11) Also, it is confusing that the text in the discussion repeatedly mentions "reverse rotation" while corresponding Fig 6 states "counter clockwise rotation".

We agree. We have removed most reference to rotation in case this was confused with actual rotary motion of the entire hexamer- now we refer to clockwise and counter-clockwise motion to describe the direction of conformational change around the hexamer. Rotation is only used when describing the axial rotation of the C2 symmetric axis within the hexamer due to nucleotide cycling. We have extensively reworded the Discussion to clarify how clockwise and counter-clockwise motion is linked to the nucleotide switch model. Two examples include:

i) in L396-400 we state: 'Conversely, one way to achieve pilus retraction would be to promote counter-clockwise motion in the CpaF motor. This could be achieved if the open' state switched to have high affinity for ATP and the open state to have low affinity for ADP thereby reversing the direction of nucleotide-induced conformational changes over the hydrolysis cycle. In support of this nucleotide switch model,....'

ii) in L408-411 we state: 'As the nucleotide binding pockets are solvent accessible in both these states, the CpaF_{ADP} structure is consistent with the nucleotide switch model with ATP and ADP binding in the open' and open states, respectively, so as to support counter-clockwise motion (Figure 6B).'

12) Also, re-wording parts of the discussion to better explain the main conclusion and also its limitations are necessary to make this paper more accessible for the non-specialist reader.

We agree and as well as reworking the Discussion as mentioned above, we now include a concluding paragraph which summarises the main conclusion of the paper. Within this paragraph as part of the limitations we discuss how other models for CpaF bifunctionality may be feasible and use the bidirectional flagellum motor as an example.

Reviewer #3 (Remarks to the Author):

In this manuscript, the authors use CryoEM to resolve the structure of CpaF, the ATPase driving the extension and retraction of Type IV-like Tad pili in *Caulobacter crescentus*. Hexameric structures of the CpaF apo form and of CpaF in the presence of AMP or AMPPNP, a non-hydrolysable analog of ATP, show a characteristic symmetric arrangement of alternate open

and closed monomer conformations, similar to what was reported for related bacterial pili ATPases and ATPases of the pili-like archaeal motor. These studies reveal atomic details of nucleotide binding to CpaF, based on which the authors postulate an ATP-dependent power stroke model in line with previous reports, where sequential ATP binding, catalysis and release drive CpaF rotation. Finally, AlphaFold predictions are used to define interaction surfaces between CpaF and the pili platform proteins CpaG and CpaH. Mutational analyses coupled to functional assays confirm that conserved residues of proposed interaction surfaces are required for Tad pili function.

Overall, the authors' work nicely resolves the structure of CpaF and its nucleotide binding conformations with reasonable resolution and convincing ligand binding information. This is the first structure of an ATPase of the T4cP subfamily, which was recently proposed to be able to rotate in both directions, representing an evolutionary ancestor of the widespread unidirectional pili ATPases. These results provide the basis for a better understanding of ATPase reversibility and the authors put forward an interesting model for bi-directional rotation. However, the experimental findings substantiating bidirectional activity of CpaF remain premature and the model entirely speculative.

We thank the reviewer for their supportive comments. The nucleotide switch model we present for CpaF bidirectional activity is consistent with the structural studies presented. However, to acknowledge that this is a model and as described previously for Reviewer 2, we have now included a concluding paragraph in the Discussion that highlights the limitations of the model and acknowledges that other mechanisms are feasible whilst discussing the bidirectional bacterial flagellum as an example. We feel our current study represents an important and significant step forward for the TadPS field and represents an exciting testable framework for future studies. Conclusively verifying the model experimentally is not trivial and we have described a future workplan (#10 above) that will test the model in a follow-on study and funding cycle.

Major points:

13) The mechanistic model of CpaF clockwise rotation for pilus extension and counterclockwise rotation for pilus retraction is based on the similarities between the P-loop disorder of the AMPPNP and ADP structures (page 13, 1st paragraph). A map of this region (P-loop, active site) of the two different structures would help readers to be able to better understand the proposed model.

Map schematics for CpaF_{AMPPNP} and CpaF_{ADP} active sites have been generated in the open, closed and open' states. This new Supplementary Figure 5, which is a very useful aid for the reader, has been referenced in the text both in the Results and Discussion.

14) CpaF mutations impacting ATPase activity have been described previously (PMID: 31897429). The observation that these mutations equally affect pili extension and retraction, has led to the hypothesis of CpaF bidirectionality. Can the authors rationalize these phenotypes based on their structural data?

Bidirectional pilus processing in the Tad pilus system motor CpaF

This is an interesting point. The main mutants described⁶ include F243L/K244R, D309N and I354C (although note that it appears incorrect numbering has been applied in Ellison et al. where these mutants are F244L/K245R, D310N and I355C <https://www.uniprot.org/uniprotkb/A0A0H3CDS2/entry#sequences>).

-F243L/K244R can be rationalised as K244R is positioned at the mouth of the nucleotide binding pocket and coordinates to the nucleotide ribose moiety. Consequently, we have included this sentence in the results L202-205: K244 is 4.0 Å from the ribose so within range for hydrogen bonding (Figure 2A and Supplementary Figure 5). Note that a K224R mutation (in combination with F243L) reduces pilus extension and retraction rate⁶ possibly by modulating nucleotide access to the binding pocket or impeding its binding.

-D309N forms part of the conserved ASP box where conserved E312 coordinates the Mg²⁺. It makes sense that D309N may therefore modulate nucleotide cycling and reduce pilus extension/retraction rates. The following sentence has been added to the legend of Supplementary Figure 5: Note that K244R (in combination with F243L) and D309N (forms part of the ASP box with E312) mutants exhibit reduced pilus extension and retraction rates⁶.

-I354C is located within the CTD fold and is harder to rationalise other than it is in relatively close proximity to Walker B E357.

15) Likewise, can the structure predict residues that are essential for bidirectionality? One would expect that such residues are conserved among bidirectional orthologs and missing in their uni-directional counterparts. If such residues can be identified, is it possible to generate unidirectional (extension only) CpaF mutants?

Intriguing idea which we followed up by aligning five CpaF, PilB and PilT homologues. This yields 25 amino acids conserved in CpaF covering many parts of the structure and whose sequence differ in both PilB and PilT. Interestingly, this alignment picked up F243 and K244 as possible candidates. These are the pair previously identified by a screen designed to select for pilus retraction deficient mutants (extension only)⁶. In this case, these mutants equally reduced the rate of both pilus extension and retraction, rather than functioning as sole retraction mutants. To verify the nucleotide switch model proposed in the paper, ideally one needs to capture a hyper-retraction form rather than an extension only mutant as the latter may preclude formation of the open' state with ATP bound. Future studies beyond the scope of this work will consolidate some of the mutants suggested by the alignments and will test them in the workplan as described above in #10.

16) The role of Mg is unclear and should be discussed by the authors. In the methods it is stated that lysis was done in a buffer containing magnesium, but it is unclear if Mg was added after this first step and could have been washed away during purification. Could this explain why Mg is found only in the closed state of CpaF but not in the Open' state in the presence of AMPPNP?

We have reviewed the purification Methods section and the final gel filtration buffer (20 mM HEPES pH7.4, 150 mM NaCl, 3 mM MgSO₄) for CpaF purification was mistakenly omitted. This has been included now in Methods. To obtain CpaF_{AMPPNP} and CpaF_{ADP} structures, purified

CpaF was incubated with a total of 5mM MgSO₄ plus desired nucleotide before vitrification. Given that Mg²⁺ was present in saturating quantities, this suggests that for CpaF_{AMPPNP} the absence of Mg²⁺ in the open and open' states was by design and was not due to its absence in the buffer. In L169-170 we now state: 'Mg²⁺ was present in the vitrification buffer indicating a requirement for specific active site geometry to support its recruitment'. Note also that the CpaF_{APO} structure has no Mg²⁺ bound in any of the states despite Mg²⁺ saturation in the buffer when vitrified.

17) And why is Mg only found in the Open' state when AMPPNP is absent? Can the authors exclude the possibility that by saturating the amounts of Mg, the ADP open' would also contain a Mg ion and that this could change the conformational state (e.g., to closed state)?

We think the reviewer is referring to the CpaF_{ADP} structure here as this is the only one where Mg²⁺ is bound in the open' state. Note that in CpaF_{ADP}, ADP and Mg²⁺ is observed in open, closed and open' states. Therefore, Mg²⁺ bound to the open' state does not promote transition to a closed state. If the reviewer is referring to CpaF_{AMPPNP}, then as outlined above, as the sample was vitrified with 5mM MgSO₄ its presence in solution is insufficient to change the conformational state. Note that the incorporation of the new Supplementary Figure 5 as suggested by this reviewer has now helped to clarify which states have Mg²⁺ bound 'at a glance'. We have also explicitly mentioned in the text that Mg²⁺ was present during vitrification for all structures.

18) What is the distance of Mg to ADP and to E357 (not shown in Figs. 2 and 3)?

Mg²⁺ to ADP to E357 distances in TADA_{ADP}: Open – 1.8 and 3.9 Å; Closed 1.8 and 3.7 Å; Open' 2.2 and 3.5 Å.

Mg²⁺ to ADP to E357 distances in TADA_{AMPPNP}: Closed 1.8 and 4.7 Å.

E357 was previously omitted from Figures 2C and 3D, and Supplementary Figure 7 (CpaF_{APO}) for clarity. However, we acknowledge the importance of this residue for the catalytic mechanism and have now added it to all figures (although in Figures 2C and 3D the distance of the sidechain to the nucleotide is still omitted so as not to overcrowd the figure. We have therefore written in the legend: 'For clarity, ASP Box E312 has been omitted as well as the distances between E357 and the nucleotide terminal phosphate (bottom row). Both residues with respective distances are shown in Supplementary Figure 5.')

19) Does any other residue participate in the coordination?

Conserved T288 in the P-Loop consensus sequence also directly coordinates with Mg²⁺ with a distance of 2Å and this is shown now in Supplementary Figure 5. ASP box E312 and Walker B E357 are loosely coordinated as they are ~4-5 Å away. These distances are shown in Supplementary Figure 5 and described in the Results. Note T288 is not shown in Figures 2C and 3D as we are sensitive to making an uninterpretable figure with too many sidechain clashes/overlaps. What we really want to showcase in these figures is how R217 and R223 are the key residues that make largescale swing movements between open and closed states to engage the nucleotide.

20) How is E357 oriented in the other subunits of CpaF-AMPPNP without Mg in comparison to the Mg bound subunits?

E357 is now shown in Figures 2C and 3D, and Supplementary Figure 7. The orientation is similar in all states with proximity to the nucleotide shifting depending on nucleotide state and whether Mg²⁺ is bound. Please also see Supplementary Figure 5.

21) The observation that CpaF-AMPPNP, CpaF-ADP and CpaF apo structures are very similar, suggests that the structural states are not driven by bound nucleotides, but rather are an inherent feature of the protein hexamer under the conditions used for CryoEM. What is the experimental evidence that open and closed states are indeed determined by the bound nucleotides?

Whilst the CpaF_{AMPPNP} and CpaF_{ADP} hexamers are similar the CpaF_{APO} is not the same. CpaF_{APO} has a similar arrangement to CpaF_{AMPPNP} and CpaF_{ADP} for the closed and open' states. However, the conformation of the open subunit NTD2 domain is significantly different (Figure 3C) resulting in an altered open state nucleotide binding pocket (Supplementary Figure 7B). The global consequence of this is that the CpaF_{APO} hexamer is more relaxed or 'rounder' than CpaF_{AMPPNP} and CpaF_{ADP} hexamers. It is clear therefore that nucleotide binding in the open state does dictate both architecture of the binding pocket and global shape of the hexamer.

To clarify this in the text we have:

-changed the title of the paragraph where CpaF_{APO} is introduced as it suggested an inappropriate level of parity between nucleotide-free and bound states. It now reads: CpaF assembles as a relaxed C2 symmetric hexamer in the absence of nucleotide

-strengthened the following sentence L298-301: However, comparison of the CpaF_{APO} and CpaF_{ADP} asymmetric units showed that whilst similar, they do not pack identically and cannot be considered equivalent to CpaF_{ADP} superposed on CpaF_{AMPPNP}.

-strengthened the summarising last line of the CpaF_{PO} structure section L310-313, it now reads: Critically though, nucleotide binding accentuates the oval-shaped C2 symmetry observed in CpaF_{AMPPNP} and CpaF_{ADP} hexamers and switches the conformation of the open state nucleotide binding pocket, with this form likely to be prevalent *in vivo* given the high nucleotide concentration in the cell.

-clarified the figure legend of Supplementary Figure 7B L865-867: For the open state active site, only the CpaF_{AMPPNP} CTD is superposed for comparison as in the absence of nucleotide the CpaF_{APO} NTD2 is rotated relative to the CpaF_{AMPPNP} NTD2 and does not align closely.

22) The details of the AlphaFold modeling of the interaction between CpaF and the platform proteins are not provided by the authors and should be included in the Methods section.

We have included details in Methods.

23) It looks like 2x CpaF, 1x CpaG and 1x CpaH was modelled. Does AF give the same interface of the two CpaF subunits as the cryo-EM structure or are they different? Are there major clashes in the model?

CpaF, CpaG and CpaH were modelled in 6:3:3 stoichiometry although Figure 5A showed 6:1:1 stoichiometry (left panel) and then 2:1:1 stoichiometry (right panel) for clarity. We have now improved the legend for Figure 5A to make this clearer. Additionally, we have created a new figure showing the 6:3:3 stoichiometry model as Supplementary Figure 9B with individual pLDDT plots in 9C (CpaF, CpaG and CpaH were 90.0, 80.1 and 82.0, respectively). The 6:3:3 model, overall pLDDT 74.9, is impressive with no clashes. AlphaFold yields similar interfaces between neighbouring CpaF subunits as in CpaF_{AMPPNP} although crucially the model is in C6 so all nucleotide binding pockets are equivalent. Each AlphaFold CpaF subunit has an RMSD C α = 2.2 Å relative to the experimentally determined CpaF_{AMPPNP} closed state subunit – this is now stated in Supplementary Figure 9B legend.

24) Do CpaG and CpaH interact?

In Supplementary Figure 9A we show the AlphaFold CpaG/CpaH heterodimer extracted from the 6:3:3 model and showcase how it has a similar fold to type 4 pilus PilB and T2SS GspF. We also separately tested modelling CpaG dimer and CpaH dimer. In both these instances a CpaG/CpaH heterodimer, PilB or GspF-like fold is not generated. Instead, an arrangement similar to that shown in Supplementary Figure 9B is generated (Figure 1 below). When CpaG or CpaH are separately modelled as a trimer they form the equivalent C3 motif as shown in Supplementary Figure 9B. AlphaFold therefore supports the CpaG/CpaH heterodimer as the functional unit.

Figure 1. AlphaFold models of CpaG and CpaH dimers. Note that PilB or GspF-like folds are not modelled unlike when CpaG is modelled with CpaH (Supplementary Figure 9A)

25) On page 14 and Figure 7, the authors discuss the stoichiometry differences of their models. However, in Figure 5, only one CpaG and one CpaH protomer are bound to a CpaF hexamer. Can the authors model 2 or 3 CpaG/H pairs to one CpaF hexamer, to investigate potential collisions? Would CpaF residues relevant for CpaG/H interaction remain at the interface?

As Supplementary Figure 9B shows CpaF:CpaG:CpaH 6:3:3 stoichiometry models with CpaF as a C6 hexamer with CpaG and CpaH forming a heterodimer arranged with C3 symmetry. With CpaF C6 symmetry there is space for CpaG:CpaH in 3:3 stoichiometry to bind without clashes. With CpaF C2 symmetry it is likely that CpaG:CpaH in 3:3 stoichiometry can still bind (although

would need to switch to C2 symmetry arrangement) as the key interface residues remain exposed. However, when the Central Loop is in the low position (open state subunit) then CpaG/CpaH will likely be blocked from binding due to the steric hindrance from the loop 8 motif above (Figure 5B). This effect likely supports dynamic binding; and is already described at L461-463.

26) Also, most CpaF mutants show reduced levels and potential degradation products (Supplementary Figure 7A) indicate that these constructs may not be physiological. Are the observed phenotypes indeed based on disrupted protein-protein interaction or are they the result of limited overall protein concentration/stability?

We shared similar concerns regarding some of the mutants including G147A/G149A, G204A/R205A/R206A, D208K, E331K/G332A/S333A/G334A and Δ E331-G334. We repurified these constructs and are satisfied that G147A/G149A, D208K, and E331K/G332A/S333A/G334A yield particles that form hexamers (Supplementary Figure 8B). However, out of caution we now note in the text that their *in vivo* expression levels and stability during *in vitro* purification were lowered stating L361-364: 'Whilst it should be cautioned that a proportion of this inhibition may be due to lowered CpaF expression *in vivo* (Supplementary Figure 8A) and stability during purification *in vitro*, the inhibition is consistent with the mutated residues playing an essential role in CpaF function, likely by interfacing with the platform proteins CpaG and CpaH'. For the G204A/R205A/R206A and Δ E331-G334 we were not satisfied by the protein stability during repurification and have removed them from the paper. These were redundant mutants anyway as both D208 and G204/R205/R206 reside on Loop 8 whilst E331K/G332A/S333A/G334A and Δ E331-G334 are in equivalent positions on the Central Loop. Their removal does not therefore change any conclusions and we should have been more judicious about including them in the first place.

27) Structural details in figures 2 and 3 are hard to grasp and color codes are confusing.

We have reviewed these figures and fiddled with various different colour codes searching for an improvement. Having taken soundings from various colleagues, in our opinion the panels are relatively clear given there are multiple different subunits that need distinguishing. The panels most open to confusion are the superpositions, particularly Figure 2B. Mindful of the reviewers concerns we have experimented having a single colour for each state but this can lead to its own problems as it is not so easy for the reader to understand that two subunits contribute to each state. Whilst acknowledging the colour codes may not be perfect, on balance we think the figures are sufficiently accessible to the reader and would like to leave them as they are please. However, to provide additional clarification for the reader for the superpositions in Figure 2B we now state in the legend: CpaF_{AMPPNP} subunits follow the colour code used in **A**.

28) I would recommend switching the order of Figs. 4 and 5 and change the text on p.10/11 accordingly. It is difficult for the reader to understand the functional assays without the interaction model having been introduced.

We agree that it was difficult to understand the functional assays for the CpaF-CpaG/H interaction model without it already being introduced. However, we do not fully support the

idea of switching Figures 4 and 5 around as this introduces other conflicts as Figure 5 is not relevant to many of the truncations/mutants such as Δ aa1-78, Δ aa1-146, and G147A/G149A. Our solution is to switch various paragraphs around so that the AlphaFold model is now introduced before the CpaF-CpaG/H interface mutants, please see L352 onwards. We feel this sufficiently resolves the issue and hope the reviewer agrees.

Minor points:

29) The type IV pili field is notorious for its lack of a common nomenclature for conserved components of the same machinery in different organisms. However, one would expect that the same nomenclature is used at least for the same species. Other reports on the Caulobacter pilus system have used a different nomenclature (CpaF for TadA and Pila for Flp1) and, to avoid confusion, I would strongly advise to adhere to this terminology.

We switched to TadA, TadB and TadC as this nomenclature is more universal in the field, we feel more intuitive, and we hoped (perhaps naively) that in time the field would converge on using this single common nomenclature rather than propagating multiple different forms. However, we fully appreciate the reviewer's point and have reverted to the Caulobacter-specific nomenclature.

30) In Supplementary Figure 7C, sample 4 shows lower distribution compared to wild type (sample 1 and 3). This should be mentioned and discussed in the main text as evidence for the functionality of N-term region. It is also interesting that there is no difference in the frequency of piliated cells (Figure 7D). Does this suggest that the N-term region is important for effective retraction, but not is affecting the extension of the pili (thus leading to less cell-body label from extra-cellularly labelled pili)? The phage assays will not contradict this result, as phages may be able to infect even the retraction is less effective. Please comment.

We thank the reviewer for highlighting this finding which we were previously overly cautious to interpret. Please see #9 above for our comment on this.

31) Page 12: "confirming a loss of pilus extension rather than just a retraction defect". It would be easier for the reader to understand this part if the observed retraction defects of other pili systems are mentioned, such as elongated pili in retraction ATPase mutant in other organisms.

We have considered this extensively and did not readily find a reference or comment that could be inserted in the text that might improve clarity. In L324-332 we carefully explain the fluorescence microscopy assay and explicitly say and already provide a reference to where pili in retraction mutants can be observed. For example: 'Addition of dye to cell culture results in direct labelling of extended cell-surface pili and also the cell body due to retraction of fluorescent pilin subunits into the cell envelope. Pilus extension mutants therefore have no fluorescent pili or cell bodies, whereas pilus retraction mutants have fluorescent pili but dark cell bodies⁶'. We believe this is sufficiently clear for readers to be powered to understand this section.

References:

1. Kachlany, S. C. *et al.* Nonspecific Adherence by *Actinobacillus actinomycetemcomitans* Requires Genes Widespread in *Bacteria* and *Archaea*. *J Bacteriol* **182**, 6169–6176 (2000).
2. Planet, P. J., Kachlany, S. C., DeSalle, R. & Figurski, D. H. Phylogeny of genes for secretion NTPases: Identification of the widespread *tadA* subfamily and development of a diagnostic key for gene classification. *Proceedings of the National Academy of Sciences* **98**, 2503–2508 (2001).
3. Clock, S. A., Planet, P. J., Perez, B. A. & Figurski, D. H. Outer Membrane Components of the Tad (Tight Adherence) Secretion of *Aggregatibacter actinomycetemcomitans*. *J Bacteriol* **190**, 980–990 (2008).
4. Planet, P. J., Kachlany, S. C., Fine, D. H., DeSalle, R. & Figurski, D. H. The Widespread Colonization Island of *Actinobacillus actinomycetemcomitans*. *Nat Genet* **34**, 193–198 (2003).
5. Tomich, M., Planet, P. J. & Figurski, D. H. The *tad* locus: postcards from the widespread colonization island. *Nat Rev Microbiol* **5**, 363–375 (2007).
6. Ellison, C. K. *et al.* A bifunctional ATPase drives *tad* pilus extension and retraction. *Sci Adv* **5**, eaay2591 (2019).
7. Lacabanne, D. *et al.* ATP Analogues for Structural Investigations: Case Studies of a DnaB Helicase and an ABC Transporter. *Molecules* **25**, 5268 (2020).
8. McCallum, M., Tammam, S., Khan, A., Burrows, L. L. & Howell, P. L. The molecular mechanism of the type IVa pilus motors. *Nat Commun* **8**, 15091 (2017).
9. McCallum, M. *et al.* Multiple conformations facilitate PilT function in the type IV pilus. *Nat Commun* **10**, 5198 (2019).
10. Satyshur, K. A. *et al.* Crystal Structures of the Pilus Retraction Motor PilT Suggest Large Domain Movements and Subunit Cooperation Drive Motility. *Structure* **15**, 363–376 (2007).

REVIEWERS' COMMENTS

Reviewer #1 (Remarks to the Author):

In this revised version of their manuscript, Hohl and co-authors have extensively addressed the comments from all three reviewers. The resulting manuscript is of high quality, and I am delighted to recommend its publication in Nature Communications.

I somewhat regret that the nomenclature was changed from the Tad/Flp nomenclature to the Cpa/Pil nomenclature, as I agree with the authors' initial choice of nomenclature, that I find a lot less confusing. However, that change was requested by reviewer 3, and I don't think it is necessary to have multiple back-and-forth changes on this very specific point.

Reviewer #2 (Remarks to the Author):

The authors addressed all of my comments very carefully. I am fully satisfied with the revised manuscript and happily recommend publication in Nature Communications.

Reviewer #3 (Remarks to the Author):

In the revised version of the manuscript, the authors have answered all of my points to my complete satisfaction.

Congratulations to a very nice piece of work.